**REPORT**

# ATG conjugation–dependent/independent mechanisms underlie lysosomal stress–induced TFEB regulation

Shiori Akayama[1,2]* , Takayuki Shima[2,3]* , Tatsuya Kaminishi[4] , Mengying Cui[4] , Jlenia Monfregola[5] , Kohei Nishino[6] , Andrea Ballabio[5,7,8,9] , Hidetaka Kosako[6] , Tamotsu Yoshimori[10] , and Shuhei Nakamura[2,3]

TFEB, a master regulator of autophagy and lysosomal biogenesis, is activated by several cellular stresses including lysosomal damage, but its underlying mechanism is unclear. TFEB activation during lysosomal damage depends on the ATG conjugation system, which mediates lipidation of ATG8 proteins. Here, we newly identify ATG conjugation–independent TFEB regulation that precedes ATG conjugation–dependent regulation, designated Modes I and II, respectively. We reveal unique regulators of TFEB in each mode: APEX1 in Mode I and CCT7 and/or TRIP6 in Mode II. APEX1 interacts with TFEB independently of the ATG conjugation system, and is required for TFEB stability, while both CCT7 and TRIP6 accumulate on lysosomes during lysosomal damage, and interact with TFEB mainly in ATG conjugation system–deficient cells, presumably blocking TFEB activation. TFEB activation by several other stresses also involves either Mode I or Mode II. Our results pave the way for a unified understanding of TFEB regulatory mechanisms from the perspective of the ATG conjugation system under a variety of cellular stresses.

## Introduction

Transcription factor EB (TFEB), a member of the MiT/TFE family, plays an essential role in maintaining cellular homeostasis by regulating autophagy and lysosomal genes bearing the Coordinated Lysosomal Expression and Regulation (CLEAR) motif (Sardiello et al., 2009; Settembre et al., 2011). A series of studies revealed that activation of TFEB function is essential to prevent the progression of several prominent diseases, including neurodegenerative diseases and metabolic disorders (Napolitano and Ballabio, 2016), and these findings have attracted much attention to the mechanisms of TFEB regulation. TFEB activity is mainly regulated by its phosphorylation and nuclear–cytoplasmic shuttling in response to changes in nutrient status. Under nutrient-rich conditions, TFEB is kept in the cytosol in an inactive state by the phosphorylation of several residues, including serine 211, in a mechanism dependent on the nutrient sensor mTORC1. During starvation, TFEB is dephosphorylated due to mTORC1 inactivation and it translocates to the nucleus, leading to the transcription activation of numerous target genes involved in autophagy and lysosomal biogenesis (Roczniak-Ferguson et al., 2012; Settembre et al., 2012). In addition to changes in nutrient status, TFEB is activated by a variety of cellular stresses, including lysosomal damage (Chauhan et al., 2016; Nakamura et al., 2020), mitochondrial damage (Nezich et al., 2015), infection (Visvikis et al., 2014), endoplasmic reticulum (ER) stress (Martina et al., 2016), DNA damage (Brady et al., 2018a), proteasome inhibition (Li et al., 2019a), oxidative stress (Martina et al., 2021), and physical exercise (Mansueto et al., 2017). Moreover, in addition to autophagy and lysosomal functions, TFEB exhibits cell- or stress-specific responses such as cell-cycle regulation (Brady et al., 2018a), pro-inflammatory signals (Brady et al., 2018b), mitochondrial biogenesis (Mansueto et al., 2017; Nezich et al., 2015), and lipid metabolism (Settembre et al., 2013). However, the molecular mechanism of TFEB regulation under these cellular stresses is less understood than those under starvation conditions.

Lysosomes are acidic organelles responsible for digesting macromolecules delivered by endocytosis and autophagy pathways. Lysosomes are often damaged by several extrinsic and intrinsic factors, such as silica, crystals, oxidative stress, and

[1]Graduate School of Frontier Biosciences, Osaka University, Suita, Japan;   [2]Department of Biochemistry, Nara Medical University, Kashihara, Japan;   [3]Center for Autophagy and Anti-Aging Research, Nara Medical University, Kashihara, Japan;   [4]Graduate School of Medicine, Osaka University, Suita, Japan;   [5]Telethon Institute of Genetics and Medicine (TIGEM), Naples, Italy;   [6]Fujii Memorial Institute of Medical Sciences, Institute of Advanced Medical Sciences, Tokushima University, Tokushima, Japan;   [7]Department of Translational Medical Sciences, Federico II University, Naples, Italy;   [8]Department of Molecular and Human Genetics, Baylor College of Medicine, Houston, TX, USA;   [9]Jan and Dan Duncan Neurological Research Institute, Texas Children's Hospital, Houston, TX, USA;   [10]Department of Beyond Cell Reborn Research, Graduate School of Medicine, The University of Osaka, Suita, Japan.

*S. Akayama and T. Shima contributed equally to this paper.   Correspondence to Shuhei Nakamura: shuhei.nakamura@naramed-u.ac.jp;   Tamotsu Yoshimori: yoshimori.tamotsu.med@osaka-u.ac.jp.

lysosomotropic drugs, leading to lysosomal membrane permeabilization and cell death (Wang et al., 2018). We and others have previously shown that induction of one form of selective autophagy, lysophagy, is critical to deal with harmful damaged lysosomes and is essential for the maintenance of cellular homeostasis (Hung et al., 2013; Maejima et al., 2013). Upon lysosomal damage, newly formed double-membrane structures called autophagosomes selectively sequester damaged lysosomes and fuse with remaining intact lysosomes. Moreover, we have shown that TFEB is activated upon lysosomal damage and its function is essential to maintain lysosomal integrity (Nakamura et al., 2020). Intriguingly, TFEB activation during lysosomal damage requires a subset of autophagy-related genes (ATG), such as ATG3, ATG5, and ATG7, collectively called the ATG conjugation system. During autophagy, this system mediates the lipidation of ATG8 proteins and its localization on the double membrane of autophagosomes. However, during lysosomal damage, the ATG conjugation system also mediates the lipidation and localization of ATG8 on the single membrane of lysosomes, and this "noncanonical" function of the ATG conjugation system is required for TFEB activation. Mechanistically, ATG8 interacts with the lysosomal calcium channel TRPML1 and mediates calcium release from lysosomes, which blocks mTORC1-dependent TFEB phosphorylation, resulting in the nuclear translocation and activation of TFEB (Nakamura et al., 2020). Importantly, these lysosomal stresses inhibit only noncanonical mTORC1 pathway: only TFEB phosphorylation is inhibited, while S6K phosphorylation is not inhibited (Napolitano et al., 2022; Napolitano et al., 2020). However, it is poorly understood why impairment of the ATG conjugation system blocks TFEB activation and how TFEB actually contributes to maintaining lysosomal integrity. Recent evidence suggests that the noncanonical function of the ATG conjugation system is also involved in TFEB activation by other cellular stresses such as mitochondrial damage and pathogen infection (Goodwin et al., 2021; Kumar et al., 2020). These observations prompted us to investigate the further mechanisms by which the ATG conjugation system regulates TFEB during lysosomal damage and its versatility under other stress conditions.

In this study, we showed that TFEB is regulated by at least two different mechanisms during lysosomal damage: ATG conjugation system–independent regulation, which was previously unidentified, and the subsequently occurring ATG conjugation system–dependent regulation; these two types of regulation are termed Mode I and Mode II, respectively. The presence of two different modes of regulation was further supported by the identification of different transcriptional programs and novel interacting proteins in each mode. Furthermore, we revealed that TFEB regulation by other cellular stressors such as oxidative stress, proteasome inhibition, mitochondrial damage, and DNA damage involves either Mode I or II.

## Results and discussion
### Identification of ATG conjugation system–independent and conjugation system–dependent TFEB regulation during lysosomal damage
To clarify the detailed mechanism of the ATG conjugation system–dependent TFEB activation during lysosomal damage,

we first performed time-lapse imaging in HeLa cells to examine the dynamics of TFEB::mNeonGreen (mNG) nuclear translocation during lysosomal damage. To induce lysosomal damage, we utilized L-leucyl-L-leucine methyl ester (LLOMe), a well-characterized lysosomotropic reagent (Thiele and Lipsky, 1990). TFEB::mNG translocated to the nucleus in wild-type (WT) cells 1 h after LLOMe treatment and remained in the nucleus even after 5 h (Fig. 1 A left, and Video 1). In contrast, ATG7 knockout (KO) cells in which lipidation of ATG8s was defective exhibited impaired TFEB::mNG nuclear translocation 5 h after LLOMe treatment, a finding consistent with our previous work (Nakamura et al., 2020) (Fig. 1 A right). Surprisingly, however, we found that TFEB::mNG transiently translocated to the nucleus 1 h after LLOMe treatment, even in ATG7 KO cells (Fig. 1 A right and Video 2). This result suggests that the ATG conjugation system, and thus ATG8 lipidation, is required for the maintenance rather than induction of TFEB nuclear translocation during lysosomal damage. Previously, we showed that the ATG conjugation system is required for lysosomal calcium release through the interaction between lipidated ATG8s on lysosomes and the lysosomal calcium channel TRPML1, which is critical for robust calcium efflux from lysosomes and subsequent TFEB nuclear translocation (Nakamura et al., 2020). To determine whether the ATG conjugation system is essential for the induction or maintenance of TFEB nuclear translocation by TRPML1-dependent calcium release, we examined the dynamics of TFEB::mNG following treatment with MK6-83, a TRPML1 agonist (Chen et al., 2014). We found that TFEB::mNG translocated to the nucleus in WT cells 1 h after MK6-83 treatment, and remained in the nucleus even after 5 h (Fig. 1 B, left, and Video 3). However, in contrast to the LLOMe condition, MK6-83 treatment failed to induce TFEB::mNG nuclear translocation in ATG7 KO cells both at 1 and at 5 h (Fig. 1 B, right, and Video 4), suggesting that the ATG conjugation system is required for the induction of TFEB nuclear translocation upon lysosomal calcium efflux.

These observations were further confirmed by western blotting. mTORC1 phosphorylation of TFEB, such as at Ser211, causes TFEB to be retained in the cytosol in an inactive state (Roczniak-Ferguson et al., 2012; Settembre et al., 2012). As a response to several types of stress, the dephosphorylation of TFEB promotes its activation and nuclear translocation. Both LLOMe and MK6-83 treatment clearly induced TFEB dephosphorylation, as demonstrated by the downshift of the TFEB band (Fig. S1, A–D). In ATG7 KO cells, consistent with live-cell imaging, the downshift of TFEB was only transiently observed at 1 h but not at 3 h after LLOMe (Fig. S1 A). In contrast, MK6-83 treatment never induced the downshift of TFEB in ATG7 KO cells (Fig. S1 B). Of note, neither LLOMe nor MK6-83 treatment largely affected the phosphorylation status of S6K, another mTORC1 substrate (Fig. S1, C and D), indicating that neither treatment caused inactivation of the canonical mTORC1 pathway, which is consistent with the recent finding that TFEB undergoes noncanonical mTORC1 regulation during lysosomal stress (Fig. S1, A and B) (Goodwin et al., 2021; Nakamura et al., 2020; Napolitano et al., 2020). Taken together, these results suggest that TFEB regulation during lysosomal damage can be largely classified according to whether the regulation is

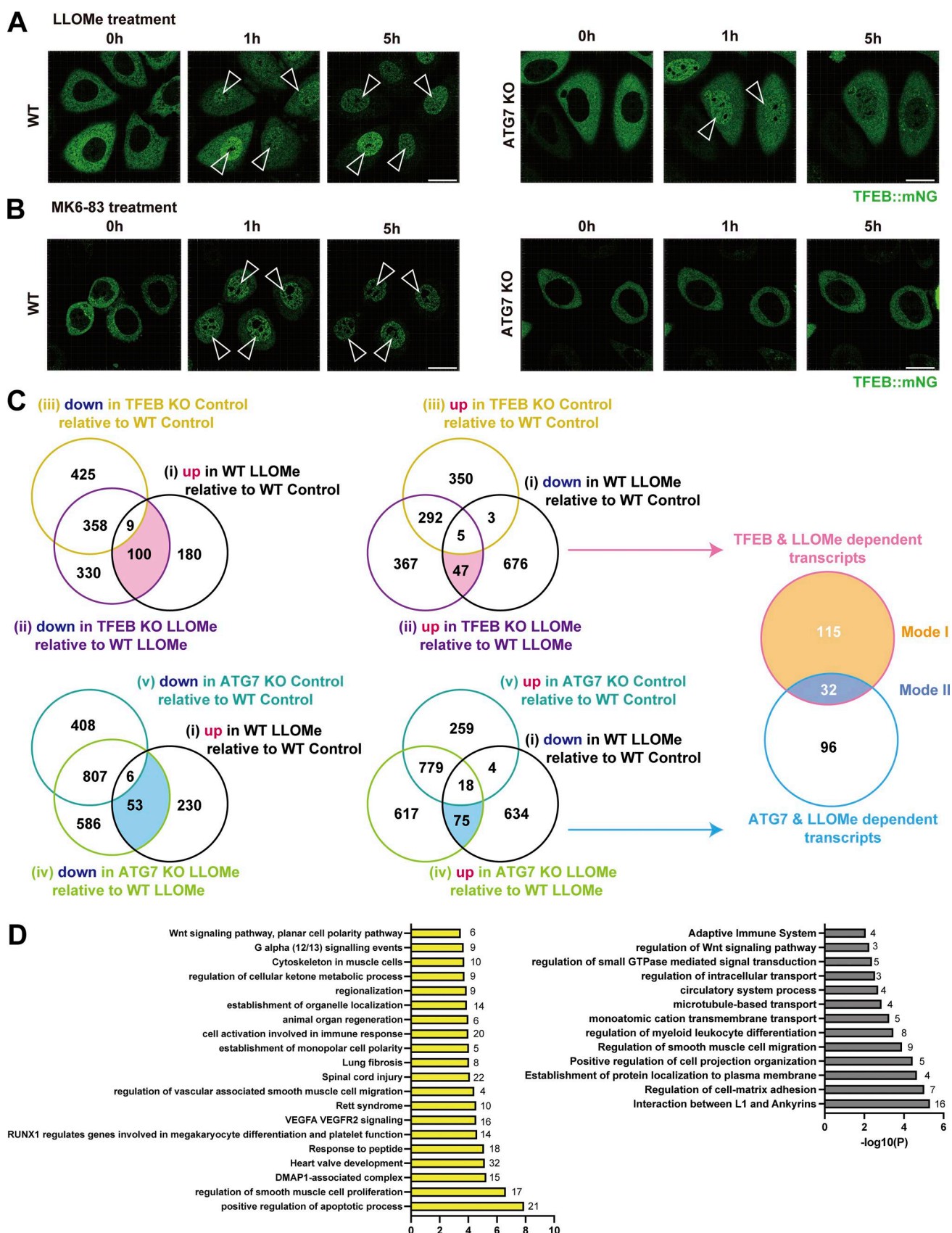

**Figure 1. Identification of both ATG conjugation system–dependent and ATG conjugation system–independent TFEB regulation during lysosomal damage. (A and B)** Representative snapshots of time-lapse imaging of TFEB::mNG in WT (left) and *ATG7* KO (right) HeLa cells. Cells were treated with 1 mM

LLOMe (A) or 100 µM MK-683 (B) for the indicated times. Arrowheads indicate nuclear-localized TFEB::mNG. **(C)** Venn diagram showing the overlap among five sets of DEGs: (1) down- or up-regulated genes in WT cells under LLOMe treatment relative to WT cells under control conditions (black circle), (2) down- or up-regulated genes in *TFEB* KO cells under LLOMe treatment relative to WT cells under LLOMe treatment (purple circle), (3) down- or up-regulated genes in *TFEB* KO cells under control conditions relative to WT cells under control conditions (yellow circle), (4) down- or up-regulated genes in *ATG7* KO cells under LLOMe treatment compared with WT cells under LLOMe treatment (yellow-green circle), and (5) down- or up-regulated genes in *ATG7* KO cells under control conditions compared with WT cells under control conditions (green circle). This analysis identified 147 genes as TFEB-dependent targets during lysosomal damage (pink circle) and 128 genes as ATG7-dependent targets during lysosomal damage (light blue circle). 115 genes were regulated by TFEB but not ATG7, corresponding to Mode I TFEB targets (orange). 32 genes were regulated by TFEB and ATG7, corresponding to Mode II TFEB targets (blue). All DEGs are listed in Table S1. **(D)** GO enrichment analysis of Mode I and Mode II TFEB target genes using Metascape. The *y*- and *x*-axes show enriched terms and enrichment with the top biological process terms plotted according to $-\log_{10}$ P value, respectively. The numbers of genes in each terms are shown. Gene lists for each GO term are shown in Table S2. Scale bars, 20 µm (A and B).

independent of (Mode I) or dependent on (Mode II) the ATG conjugation system, respectively.

To examine whether the two types of TFEB regulation result in different transcriptional outcomes, we conducted RNA sequencing (RNA-seq) analysis using WT, *ATG7* KO, and *TFEB* KO cells with or without LLOMe treatment for 1 h. We first extracted differentially expressed genes (DEGs) defined by P < 0.05 (quasi-likelihood F-test).

First, to identify lysosomal damage–induced TFEB-dependent targets, we determined the intersection between (1) up-regulated genes in WT cells under LLOMe treatment relative to WT under control conditions, and (2) down-regulated genes in *TFEB* KO cells under LLOMe treatment relative to WT cells under LLOMe treatment. We also examined the intersection of the inverse scenario, i.e., down- and up-regulated genes, respectively (Fig. 1 C and Table S1). Furthermore, to identify TFEB targets only under lysosomal damage, we removed (3) genes in *TFEB* KO cells that were down- or up-regulated under control conditions compared with WT cells under control conditions. This analysis identified 147 genes as lysosomal damage–induced TFEB-dependent targets (Fig. 1 C and Table S1). We noticed that only three genes, specifically *IGFR2*, *LIN52*, and *SPTBN1*, overlapped between our 147 lysosomal damage–induced TFEB targets and 471 previously described starvation-induced TFEB targets (Fig. S1 E) (Palmieri et al., 2011), implying that TFEB regulates different sets of downstream genes depending on the cellular context. In parallel, to extract ATG7- and lysosomal damage–dependent genes, we identified the intersection between (1) genes that were up- or down-regulated in WT cells under LLOMe treatment relative to WT cells under control conditions, and (4) genes that were down- or up-regulated in *ATG7* KO cells under LLOMe treatment relative to WT cells under LLOMe treatment (Fig. 1 C and Table S1). To again remove the genes constitutively regulated by ATG7 from the intersection, we eliminated (5) genes that were down- or up-regulated in *ATG7* KO cells under control conditions compared with WT under control conditions, resulting in the identification of 128 ATG7- and lysosomal damage–dependent genes (Fig. 1 C and Table S1).

A comparison of the 147 and 128 genes revealed that 115 genes were regulated by TFEB but not by ATG7, corresponding to Mode I TFEB downstream targets (TFEB and lysosomal damage but ATG7-independent genes) (Fig. 1 C). On the other hand, the 32 genes were under the control of both TFEB and ATG7, indicating that they are Mode II TFEB downstream targets (TFEB, lysosomal damage, ATG7-dependent genes) (Fig. 1 C and Table S1).

Remarkably, gene ontology (GO) analysis of each gene set using Metascape (Zhou et al., 2019) revealed nonoverlapping GO terms, such as "positive regulation of apoptotic process" specific to Mode I, and "interaction between L1 and ankyrins" specific to Mode II, suggesting that the two modes of TFEB activation in response to lysosomal damage indicate the presence of distinct biological pathways that regulate different sets of downstream genes (Fig. 1 D and Table S2). Our quantitative PCR (qPCR) analysis confirmed that a Mode I target, IL-6, was indeed regulated by TFEB-dependent and ATG7-independent manner, while a Mode II target, CCL21, was regulated by both TFEB- and ATG7-dependent manner (Fig. S1, F and G).

### Identification of Mode I and II novel regulators of TFEB during lysosomal damage

To further elucidate the mechanism of TFEB regulation in the two modes, we used a proteomics approach to identify novel TFEB-interacting proteins. Although the complete separation of two regulatory modes is difficult, our time-lapse analysis suggests that Mode I regulation mainly starts within 1 h after LLOMe treatment, while Mode II regulation becomes apparent 1 h after LLOMe. Thus, WT and *ATG7* KO cells stably expressing TFEB::mNG treated with or without LLOMe were collected at different time points (0, 1, or 3 h), and each sample was co-immunoprecipitated with an anti-mNG nanobody followed by mass spectrometry (MS) analysis (Fig. S2 A). We hypothesized that novel TFEB-interacting proteins that work as a positive regulator of TFEB and are involved in ATG conjugation system–independent regulation, Mode I, should be enriched both in WT cells (Fig. S2 B, green square) and in *ATG7* KO cells (Fig. S2 C, red square) at 1 h after LLOMe treatment compared with non-treated control cells. Our time-lapse analysis indicates that around 3 h after LLOMe, most of TFEB starts to go back to the cytosol in *ATG7* KO cells, while TFEB remains in the nucleus in WT. Therefore, we hypothesize that at this time point (3 h), interacting candidates should be enriched more in WT (TFEB mainly in nucleus) compared with *ATG7* KO (Fig. S2 D, blue square). We found that 52 candidates met these criteria when the cutoff for protein enrichment was set to a $\log_2$ fold change of >0.4 (Fig. S2 E). After excluding several abundant proteins such as ribonucleoproteins, we focused on 20 proteins as Mode I candidates. To identify novel interacting proteins essential for Mode I TFEB regulation, we knocked down each of the 20 candidates to determine those whose depletion abolished TFEB::mNG nuclear translocation after 1-h LLOMe treatment in

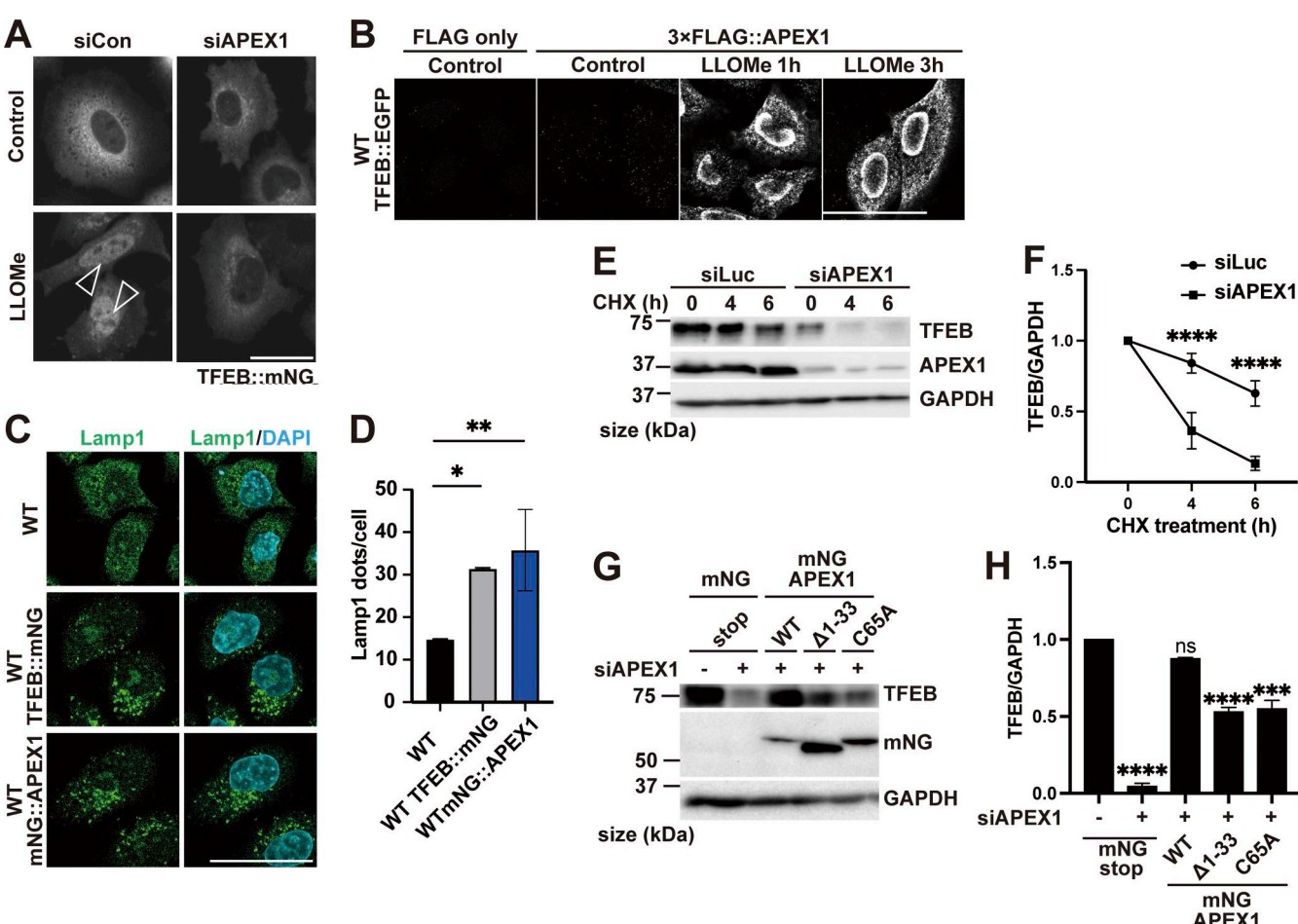

Figure 2. **Identification of a Mode I regulator of TFEB during lysosomal damage. (A)** Representative fluorescence images of HeLa cells stably expressing TFEB::mNG transfected with scrambled siRNA (si Con) or si*APEX1*. Cells were either untreated (control) or treated with 1 mM LLOMe for 1 h. Arrowheads indicate nuclear-localized TFEB::mNG. **(B)** Results of the PLA using HeLa cells stably expressing TFEB::EGFP. The indicated plasmids were transiently transfected and were either nontreated (control) or treated with 1 mM LLOMe for 1 h or for 3 h followed by the PLA procedure. **(C)** Representative fluorescence images of lysosomes stained with an antibody against LAMP1 (green) in WT HeLa cells or WT HeLa cells stably expressing TFEB::mNG or mNG::APEX1 under nutrient conditions. **(D)** Quantification of the image data shown in C (n = 3 biologically independent samples). **(E)** Representative immunoblots of TFEB, APEX1, and GAPDH transfected with siLuc or si*APEX1*. Cells were treated with cycloheximide for the indicated times (0 h: untreated control). **(F)** Quantification of immunoblot data shown in E (n = 3 biologically independent samples). **(G)** Representative immunoblots of TFEB, mNG, and GAPDH transfected with mNG:: STOP or mNG::APEX1 (KD-resistant) WT or mNG::APEX1 mutants (KD-resistant) in *APEX1* KD condition. **(H)** Quantification of immunoblot data shown in G (n = 3 biologically independent samples). **(A–C)** Scale bars, 50 μm. **(D–H)** P values were determined by one-way ANOVA with Tukey's multiple comparison test; *P < 0.05, **P < 0.01, ***P < 0.001, ****P < 0.001. ANOVA, analysis of variance. siLuc, siLuciferase. Source data are available for this figure: SourceData F2.

WT cells. This screening identified APEX1 (apurinic/apyrimidinic endodeoxyribonuclease 1), whose knockdown abolished TFEB nuclear localization, as a novel TFEB regulator (Fig. 2 A and Fig. S3 A). APEX1 is involved in the base excision repair pathway and in the redox-dependent regulation of several transcription factors, including NF-κB, Fos, Jun, and p53 (Gaiddon et al., 1999; Nishi et al., 2002; Xanthoudakis and Curran, 1992), but the role of APEX1 in TFEB regulation has not been studied. We found that *APEX1* knockdown impaired TFEB dephosphorylation as revealed by blunted reduction of phospho-Ser211 TFEB band both at 1 h and at 3 h after LLOMe (Fig. S3, B and C). Consistent with the MS result, a proximity ligation assay (PLA) revealed that 3×FLAG::APEX1 indeed interacted with TFEB::EGFP only under the condition of LLOMe treatment and this interaction persists even after 3-h LLOMe treatment (Fig. 2 B). Importantly, the PLA revealed the interaction between 3×FLAG::APEX1 and TFEB::

EGFP, also in the *ATG7* KO background (Fig. S3 D), suggesting that APEX1 is required for TFEB activation independently of the ATG conjugation system and is thus a bona fide Mode I regulator. Importantly, while other members of the MiTF family such as TFE3 and MITF translocated to nucleus in response to lysosomal damage (Fig. S3, E and F), PLA analysis revealed that neither TFE3 nor MITF interacts with APEX1 (Fig. S3 G). Consistent with a previous report (Li and Wilson, 2014), APEX1 was localized on the nucleus and we revealed that this localization was not affected by LLOMe treatment (Fig. S3 H). These data imply that the interaction between APEX1 and TFEB occurs in the nucleus under lysosomal damage conditions. We observed that *APEX1* knockdown reduced the TFEB protein level (Fig. S3 B). qPCR analysis revealed that among members of the MiTF family including TFEB, TFE3, and MITF, *APEX1* knockdown specifically reduced levels of TFEB transcripts (Fig. S4 A). Conversely, APEX1

overexpression was sufficient to increase the TFEB protein level (Fig. S4 B), the number of lysosomes (Fig. 2, C and D), and CLEAR-element–containing luciferase reporter activity under lysosomal damage (Fig. S4 C). Considering TFEB activates its own transcription (Settembre et al., 2013), we speculate that APEX1 binding to TFEB might be essential for the transcription of TFEB and the maintenance of its protein stability. Indeed, a cycloheximide chase experiment revealed that *APEX1* knockdown cells showed faster TFEB degradation compared with control knockdown cells (Fig. 2, E and F). Proteasome inhibition did not rescue protein levels of TFEB in *APEX1* knockdown cells (Fig. S4, D and E), suggesting that proteasomal degradation pathways are not involved in TFEB degradation due to the lack of APEX1. Although we examined the possible involvement of lysosomal degradation pathways in TFEB regulation, treatment with the lysosome inhibitor chloroquine for at least 4 h did not affect TFEB levels (Fig. S4, F and G). It has been shown that APEX1 plays a role in double-strand break (DSB) repair of DNA (Ströbel et al., 2017), but knockdown of *APEX1* alone did not show obvious DSB as revealed by the lack of γH2AX foci in our experimental condition (Fig. S4, H–J). Since the nuclear localization and/or the redox activity of APEX1 affects its activity (Georgiadis et al., 2008; Oliveira et al., 2022), we asked whether these domains are required for both the interaction and the stability of TFEB. We found that neither the nuclear localization signal depleted *APEX1* mutant (Δ1-33) nor the redox inactive *APEX1* mutant (C65A) interacted with TFEB as revealed by the PLA (Fig. S4, K–M). Furthermore, reintroduction of these siRNA-resistant *APEX1* mutants to an *APEX1* knockdown cell did not fully rescue the expression level of TFEB (Fig. 2, G and H). Collectively, these results indicate that APEX1 is a novel TFEB-interacting protein that plays an essential role in Mode I regulation of TFEB by modulating its stability and nuclear translocation. This regulation is dependent on both nuclear localization and redox activity of APEX1, but independent of proteasomes and the lysosomal degradation system.

Next, to identify novel factors in the ATG conjugation system–dependent regulation of TFEB, Mode II, we focused on TFEB-interacting proteins enriched in *ATG7* KO cells relative to WT cells at 3 h after LLOMe treatment, since we speculated that their physical interaction with TFEB might result in the defective TFEB nuclear translocation observed in *ATG7* KO cells. The cutoff for protein enrichment was set to a log$_2$ fold change of less than –0.5, and we identified 93 proteins (Fig. S5 A, red square). After excluding several abundant proteins, we focused on 26 proteins as Mode II candidates. We hypothesized that some of these candidates interact with TFEB, preventing nuclear translocation in the absence of lipidated ATG8s in *ATG7* KO cells, and that knockdown of the candidates might rescue TFEB nuclear localization. Indeed, knockdown of *CCT7* (chaperonin-containing TCP1 subunit 7), *TRIP6* (thyroid hormone receptor interactor 6), and *CNBP* (CCHC-type zinc finger nucleic acid–binding protein) among the candidates significantly rescued TFEB nuclear translocation in *ATG7* KO cells (Fig. 3 A; and Fig. S5 B and E). Since MK6-83 induced TFEB nuclear localization solely dependent on the ATG conjugation system and thus Mode II (Fig. 1 B), we also conducted interactome analysis of TFEB::mNG in WT

and *ATG7* KO cells after MK6-83 treatment and found that the same candidates, TRIP6, CCT7, and CNBP, were also enriched in *ATG7* KO cells compared with WT cells (Fig. S5, C and D). In contrast, APEX1 did not show up as a TFEB-interacting candidate upon MK6-83 in WT cells, when the cutoff for protein enrichment was set to a log$_2$ fold change of >0.4. Co-immunoprecipitation experiments revealed that 3×FLAG::CCT7 and 3×FLAG::TRIP6, but not 3×FLAG::CNBP, interacted with TFEB::mNG in *ATG7* KO cells with or without LLOMe treatment (Fig. 3, B and C; and Fig. S5 F). Interestingly, both CCT7 and TRIP6 were distributed throughout each cell under steady-state conditions and starvation conditions, but became colocalized with lysosomes in *ATG7* KO cells upon lysosomal damage (Fig. 3, D–G; and Fig. S5, G and H). Based on these results, we concluded that CCT7 and TRIP6 are critical Mode II regulators that inhibit TFEB nuclear translocation in the absence of the ATG conjugation system, presumably through their interactions with TFEB in *ATG7* KO cells.

A previous study suggests that noncanonical lipidation of GABARAP sequesters FLCN and disrupts its GAP activity toward RagC/D, resulting in dephosphorylation and activation of TFEB (Goodwin et al., 2021). Consistent with this finding, when *RagC/D* or *FLCN* was knocked down, TFEB translocated to the nucleus in ATG7 KO cells in response to lysosomal damage (Fig. S6, A and B), confirming that ATG conjugation system–dependent TFEB activation during lysosomal damage involves the FLCN-RagC/D-mTORC1 axis. The potential crosstalk between this axis and the function of CCT7/TRIP6 was then investigated by checking the impact of *TRIP6* and *CCT7* knockdown on the mTORC1-dependent Ser211 phosphorylation of TFEB. We observed knockdown of *TRIP6* or *CCT7* in *ATG7* KO cells significantly reduced phosphorylation of Ser211 on TFEB compared with control knockdown (siLuc) in *ATG7* KO cells (Fig. 3, H and I). In contrast, although TFEB is reported to be regulated by AMPK (Malik et al., 2023), knockdown of neither *TRIP6* nor *CCT7* affected the phosphorylation status of AMPK (Fig. S6 C). These results indicate that knockdown of either *TRIP6* or *CCT7* rescues TFEB nuclear localization in *ATG7* KO cells through mTORC1 activity, but independently of AMPK signaling. To understand the functional significance of TFEB inhibition by TRIP6/CCT7, we monitored TFEB activity during the recovery phase after the lysosomal damage in WT cells. Consistent with our previous findings (Nakamura et al., 2020), at 3 h after LLOMe wash-off, TFEB was still activated as revealed by downshift and dephosphorylation on Ser211 of TFEB in control knockdown cells (Fig. 3 J). However, at 18 h after the LLOMe wash-off, TFEB becomes inactivated as revealed by uppershift and phosphorylation on Ser211 of TFEB in these cells. In contrast, inactivation of TFEB was blunted in *CCT7* or *TRIP6* knockdown cells (Fig. 3 J). These data suggest that both CCT7 and TRIP6 function as negative regulators to play a role in terminating TFEB activation at the certain periods of time after the lysosomal damage in WT cells.

### Cellular stress–induced TFEB regulation involves either Mode I or II

In addition to lysosomal damage, various other stresses such as mitochondrial stress (Goodwin et al., 2021; Nezich et al., 2015),

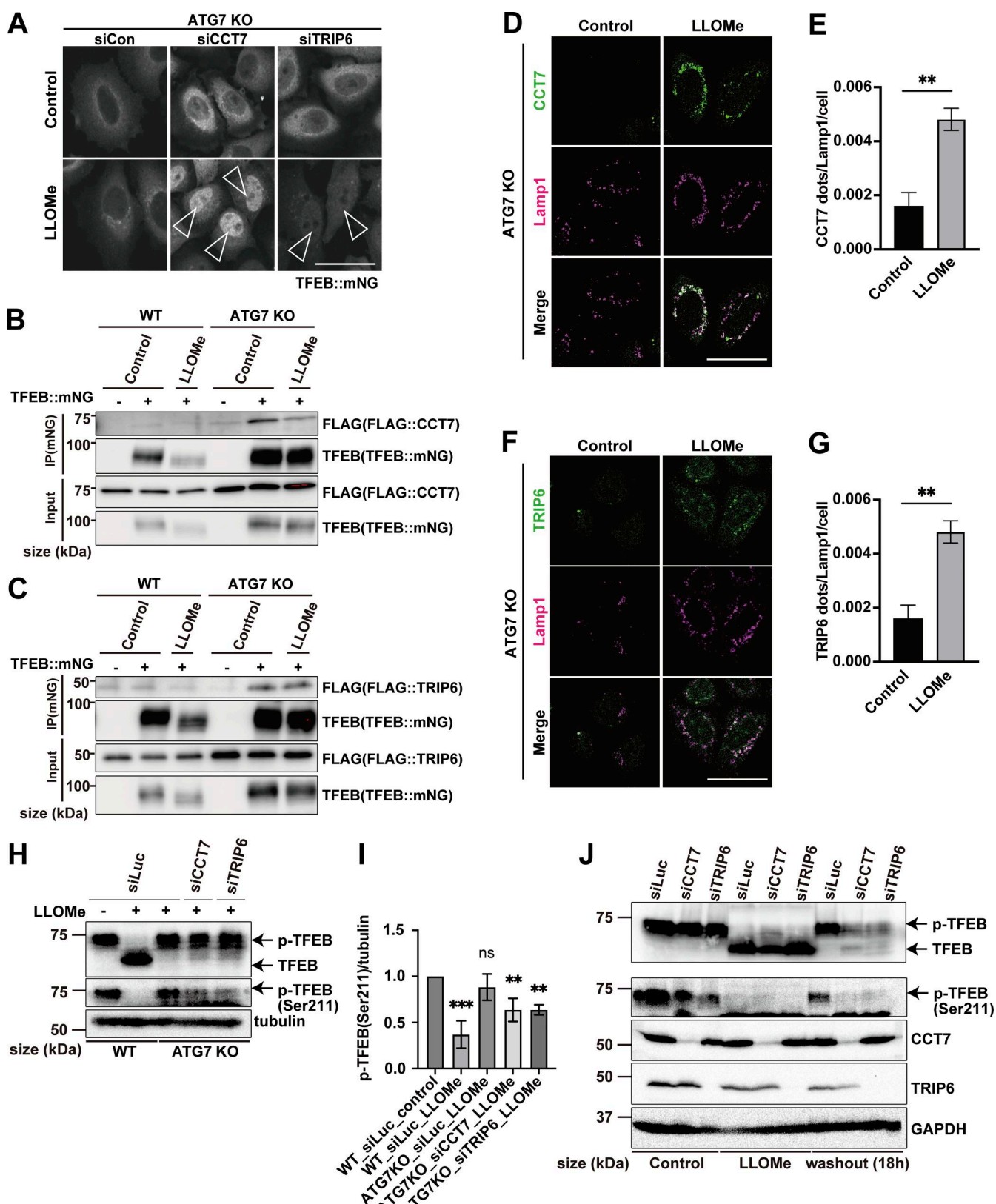

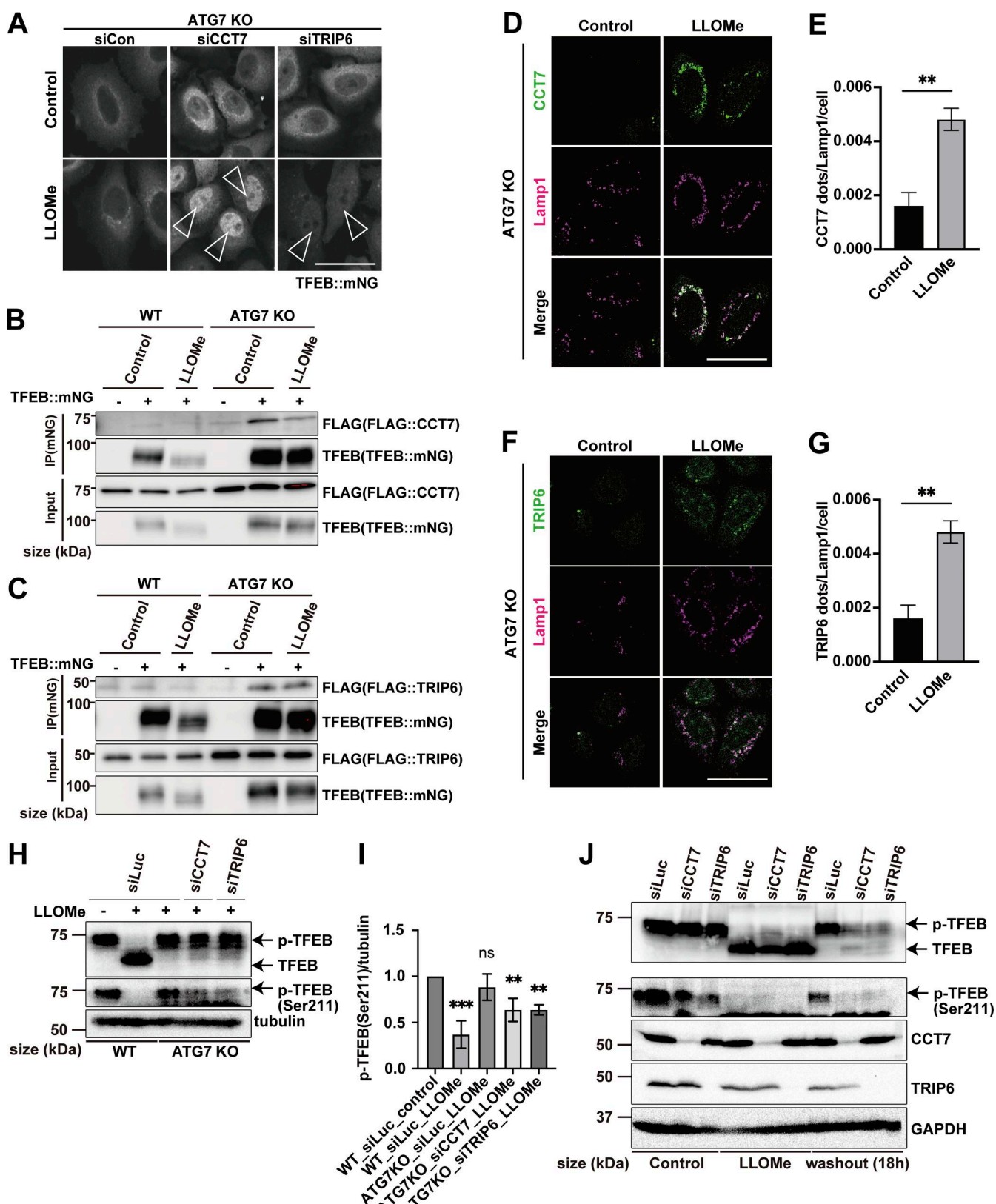

Figure 3. **Identification of Mode II regulators of TFEB during lysosomal damage. (A)** Representative fluorescence images of *ATG7* KO cells stably expressing TFEB::mNG and transfected with scrambled siRNA (siCon), si*CCT7*, and si*TRIP6*. Cells were either untreated (control) or treated with 1 mM LLOMe for 3 h. Arrowheads indicate nuclear-localized TFEB::mNG. **(B and C)** Representative immunoblots of FLAG and TFEB in WT or *ATG7* KO HeLa cells with or without stable TFEB::mNG expression. FLAG::CCT7 (B) or FLAG::TRIP6 (C) was transiently transfected and then treated or not treated (control) with 1 mM LLOMe for 3 h, followed by immunoprecipitation with an antibody against mNG. **(D and F)** Representative fluorescence images of HeLa cells stably expressing mNG::CCT7

(D) or mNG::TRIP6 (F) (green). Lysosomes were stained with an antibody against LAMP1 (magenta). Cells were either treated or not treated (control) with 1 mM LLOMe for 3 h. **(E and G)** Quantification of immunofluorescence data shown in D or G, respectively (n = 3 biologically independent samples). **(H)** Representative immunoblots of TFEB, p-TFEB (Ser211), and tubulin under control conditions and 1 mM LLOMe treatment for 3 h in WT and ATG7 KO HeLa cells. Cells were transfected with siLuc, si*CCT7*, and si*TRIP6*. **(I)** Quantification of image data of p-TFEB (Ser211)/tubulin shown in H (n = 3 biologically independent samples). **(J)** Representative immunoblots of TFEB, p-TFEB (Ser211), CCT7, TRIP6, and GAPDH in WT HeLa cells under control and 3 or 18 h after 1 mM LLOMe treatment conditions. Cells were transfected with siLuc, si*CCT7*, and si*TRIP6*. **(A, D, and F)** Scale bars, 50 μm. **(E and G)** P values were determined by a t test; **P < 0.01. **(I)** P values were determined by ANOVA with Tukey's multiple comparison test; **P < 0.01, ***P < 0.005. Source data are available for this figure: SourceData F3.

DNA damage (Brady et al., 2018a), proteasome inhibition (Li et al., 2019a), and oxidative stress (Martina et al., 2021) have been reported to induce TFEB activation. Thus, we next examined which of the two modes of TFEB regulation during lysosomal damage might participate under other cellular stress conditions. Consistent with previous reports (Goodwin et al., 2021; Nezich et al., 2015), WT cells and one of core autophagy genes, *ATG13* KO cells expressing Myc::Parkin, exhibited translocation of TFEB::mNG to the nucleus upon exposure to several mitochondrial stressors, including valinomycin (Videos 5) and oligomycin/antimycin A (O/A) (Fig. 4, A and B), while this translocation was impaired in *ATG7* KO cells (Fig. 4, A and B; and Video 6). These results suggest that the ATG conjugation system, but not autophagy itself, is required for TFEB activation under mitochondrial stress. Similarly, TFEB::mNG nuclear translocation was induced by etoposide treatment, which generates DNA DSBs in WT cells and in cells with KO of another core autophagy gene, *FIP200*, but was abolished in *ATG7* KO cells (Fig. 4, C and D). In contrast, we found that arsenite, which induces oxidative stress, and the proteasome inhibitor MG132 induced TFEB::mNG nuclear translocation in both WT cells and autophagy-deficient *FIP200* KO and *ATG7* KO cells, suggesting that the ATG conjugation system is unnecessary for TFEB activation under these stress conditions (Fig. 4, E–H). These results suggest that functions of the ATG conjugation system that are independent of its role in autophagy are required for TFEB activation induced by mitochondrial stress, DNA damage, or lysosomal stress, but not for that induced by oxidative stress or proteasome inhibition. In line with these results, we found that noncanonical lipidation of LC3B, a member of the ATG8 family, was significantly elevated under mitochondrial stress, DNA damage, lysosomal stress, or TRPML1 agonist MK6-83 treatment but not under arsenite or MG132 treatment in autophagy-deficient *ATG13* or *FIP200* KO cells (Fig. 4, I and J; and Fig. S7 A).

We further examined whether cellular stress–induced TFEB activation involved APEX1 as a Mode I regulator, and CCT7 and/or TRIP6 as Mode II regulators. Remarkably, consistent with the requirements of the ATG conjugation system, *TRIP6* knockdown reversed the defective TFEB::mNG nuclear translocation in *ATG7* KO cells under the mitochondrial stress or etoposide treatment, respectively (Fig. 5, A, B, D, and E). Of note, MK6-83 treatment, which induces lysosomal calcium efflux, also caused this defective translocation in *ATG7* KO cells and it was rescued by *CCT7* knockdown (Fig. 5, G and H). Importantly, the requirement of either *TRIP6* or *CCT7* knockdown to rescue TFEB nuclear localization under these stresses was well matched with the change in their localization. Under the mitochondrial stress and etoposide treatment, TRIP6 but not CCT7 forms dots, while CCT7 but not TRIP6 forms puncta upon MK6-83 treatment (Fig. 5, C, F, and

I; and Fig. S7, B–D). These data imply that the different dependence of CCT7 or TRIP6 is regulated at the level of their localization likely on lysosomes.

We further found that *APEX1* knockdown abolished either arsenite- or MG132-induced TFEB::mNG nuclear translocation, respectively (Fig. 5, J and K). On the contrary, *APEX1* knockdown did not abolish TFEB::mNG nuclear translocation under either O/A, etoposide, or MK6-83 treatment, respectively (Fig. S7, E–J). Taken together, our results revealed that the mechanism of cellular stress–induced TFEB activation can be classified into either the APEX1-mediated Mode I–dependent or the TRIP6/CCT7-involved Mode II–dependent regulation.

In our previous study, we showed that TFEB activation during lysosomal damage depends on the ATG conjugation system, which mediates lipidation of the ATG8 protein (Nakamura et al., 2020). In this study, time-lapse imaging led to the new discovery of ATG conjugation system–independent TFEB regulation, which precedes ATG conjugation system–dependent regulation; these two regulatory systems are designated as Modes I and II, respectively. The presence of these two systems was further supported by the identification of both TFEB-dependent transcriptomes and ATG7-dependent or ATG7-independent transcriptomes during lysosomal damage. Furthermore, our proteomics analysis and subsequent functional screening identified APEX1 as a unique Mode I regulator of TFEB. APEX1 interacted with TFEB in the nucleus only under lysosomal damage conditions, and was critical for the maintenance of TFEB stability and its nuclear translocation. We also identified CCT7 and TRIP6 as novel Mode II TFEB regulators, both of which were localized on lysosomes and interacted with TFEB under lysosomal damage conditions, presumably by blocking TFEB nuclear translocation by affecting mTORC1 activity toward TFEB in the absence of lipidated ATG8s. We also revealed that both CCT7 and TRIP6 function as negative regulators to terminate TFEB activation after the lysosomal damage in WT cells. Remarkably, the mechanism of TFEB activation induced by other cellular stressors such as mitochondrial stress, DNA damage, lysosomal calcium release, proteasome inhibition, and oxidative stress relied on either APEX1-dependent Mode I or TRIP6/CCT7-involved Mode II regulation, suggesting that these two modes are widely utilized to regulate TFEB.

Our RNA-seq data showed that TFEB-dependent transcriptional changes during lysosomal damage were different from those during starvation, again implying the presence of cellular stress–dependent targets of the MiTF family (i.e., unique targets of TFEB and TFE3 have been reported under lipopolysaccharide treatment, ER stress, inflammation, and DNA damage [Brady et al., 2018a; Martina et al., 2016; Pastore et al., 2016]).

Intriguingly, our data indicated that in Mode I, 9 of 15 TFEB targets, including nuclear receptor 4A1 (NR4A1), are involved in

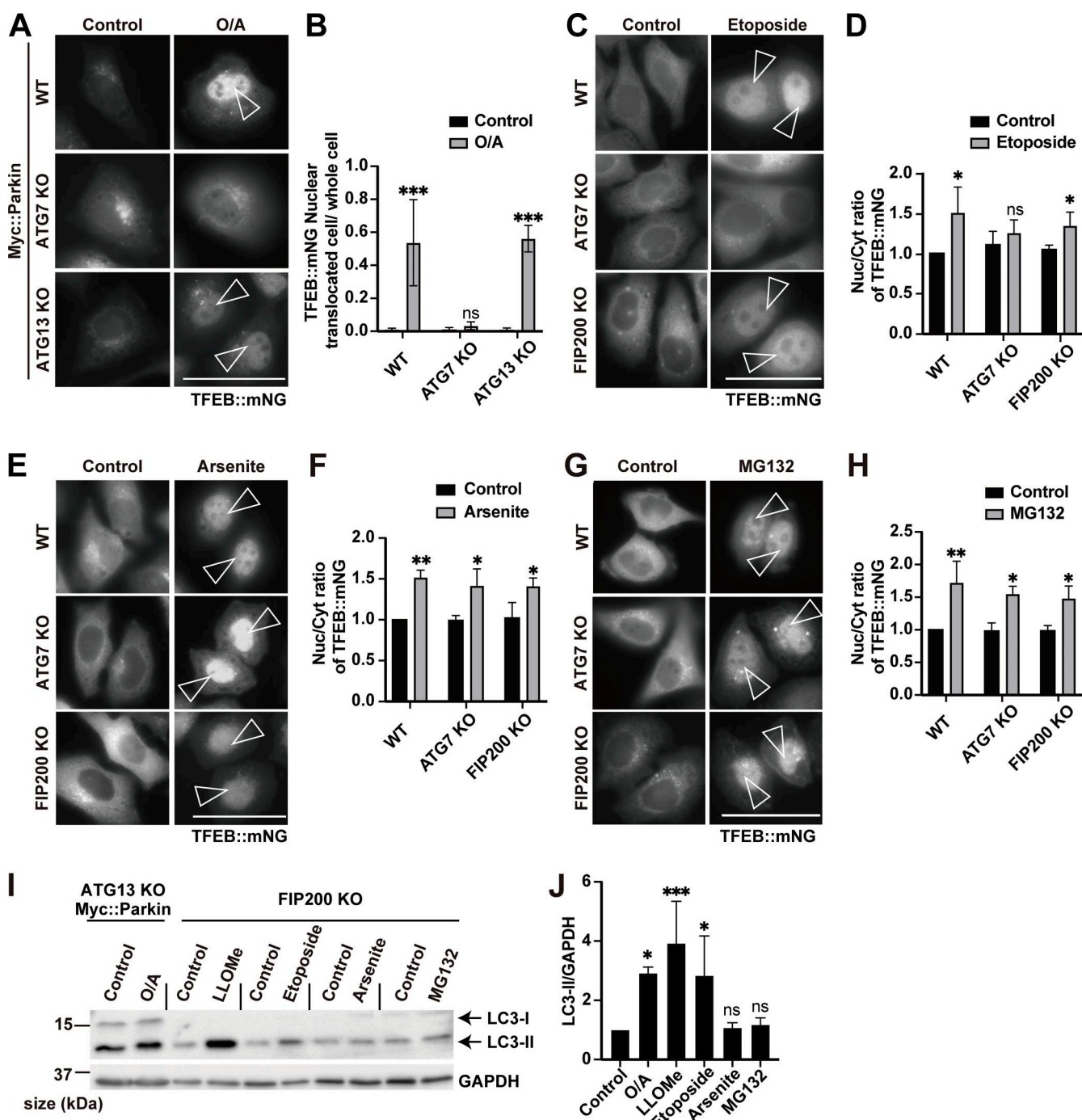

Figure 4. **Various cellular stressors induce TFEB activation in either an ATG conjugation system–independent or an ATG conjugation system–dependent manner. (A)** Representative fluorescence images of WT, *ATG7* KO, and *ATG13* KO HeLa cells stably expressing TFEB::mNG and Myc::Parkin, either untreated (control) or treated with O/A for 3 h. **(B)** Quantification of the image data shown in A (*n* = 3 biologically independent samples). **(C)** Representative fluorescence images of WT, *ATG7* KO, and *FIP200* KO HeLa cells stably expressing TFEB::mNG, either untreated (control) or treated with etoposide for 24 h. **(D)** Quantification of the image data shown in C (*n* = 3 biologically independent samples). **(E)** Representative fluorescence images of WT, *ATG7* KO, and *FIP200* KO HeLa cells stably expressing TFEB::mNG, either untreated (control) or treated with arsenite for 6 h. **(F)** Quantification of the image data shown in E (*n* = 3 biologically independent samples). **(G)** Representative fluorescence images of WT, *ATG7* KO, and *FIP200* KO HeLa cells stably expressing TFEB::mNG, either untreated (control) or treated with MG132 for 9 h. **(H)** Quantification of image data shown in G (*n* = 3 biologically independent samples). **(I)** Representative immunoblots of LC3B and GAPDH in *ATG13* KO HeLa cells stably expressing Myc::Parkin or in *FIP200* KO HeLa cells. Cells were either untreated (control) or treated with the indicated reagents. **(J)** Quantification of the image data shown in I (*n* = 3 biologically independent samples). The value of "Control" was defined as 1, and relative values compared with each control were calculated. **(A, C, E, and G)** Scale bars, 50 μm. **(A, C, E, and G)** Arrowheads indicate nuclear-localized TFEB::mNG. **(B, D, F, H, and J)** P values were determined by ANOVA with Tukey's multiple comparison test; *P < 0.05, **P < 0.01, ***P < 0.005. Source data are available for this figure: SourceData F4.

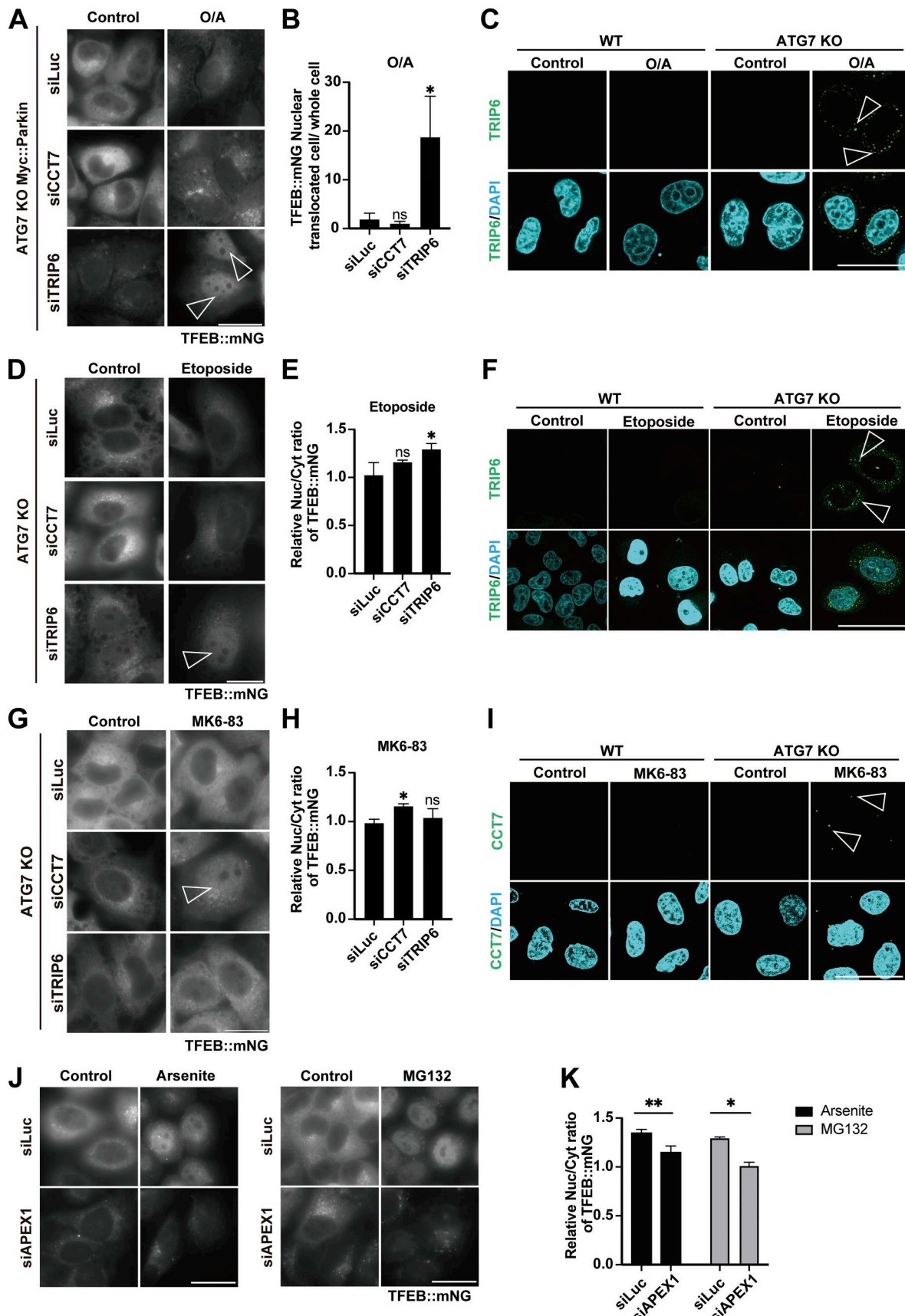

Figure 5. **TFEB activation by several cellular stressors involves either Mode I or Mode II. (A)** Representative fluorescence images of *ATG7* KO HeLa cells stably expressing TFEB::mNG and Myc::Parkin, either untreated (control) or treated with O/A for 3 h. Cells were transfected with siLuc, si*CCT7*, or si*TRIP6*.

**(B)** Quantification of the image data shown in A (*n* = 3 biologically independent samples). **(C)** Representative fluorescence images of WT or *ATG7* KO HeLa cells transiently expressing mNG::TRIP6, either untreated (control) or treated with O/A for 3 h. **(D)** Representative fluorescence images of *ATG7* KO HeLa cells stably expressing TFEB::mNG, either untreated (control) or treated with etoposide for 24 h. Cells were transfected with siLuc, si*CCT7*, or si*TRIP6*. **(E)** Quantification of the image data shown in D (*n* = 3 biologically independent samples). **(F)** Representative fluorescence images of WT or *ATG7* KO HeLa cells transiently expressing mNG::TRIP6, either untreated (control) or treated with etoposide for 24 h. **(G)** Representative fluorescence images of *ATG7* KO HeLa cells stably expressing TFEB::mNG, either untreated (control) or treated with MK6-83 for 3 h. Cells were transfected with siLuc, si*CCT7*, or si*TRIP6*. **(H)** Quantification of the image data shown in G (*n* = 3 biologically independent samples). **(I)** Representative fluorescence images of WT or ATG7 KO HeLa cells transiently expressing mNG::CCT7, either untreated (control) or treated with MK6-83 for 3 h. **(J)** Representative fluorescence images of WT HeLa cells stably expressing TFEB::mNG, either untreated (control) or treated with arsenite for 6 h or MG132 for 9 h. Cells were transfected with siLuc or si*APEX1*. **(K)** Quantification of image data shown in J (*n* = 3 biologically independent samples). **(A, C, D, F, G, I, and J)** Scale bars, 50 μm. **(A, D, and G)** Arrowheads indicate nuclear-localized TFEB::mNG. **(C, F, and I)** Arrowheads indicate TRIP6/CCT7 dots. **(B, E, H, and K)** Relative value of the TFEB nuclear translocation rate under drug treatment conditions compared with that under basal (control) conditions. P values were determined by ANOVA with Tukey's multiple comparison test (B, E, and H) or *t* test (K); *P < 0.05, **P < 0.01.

pathways related to controlling cell death, such as ferroptosis (Gao et al., 2018; Ye et al., 2021). Ferroptosis is one of the programmed cell death caused by reactive oxygen species (ROS) accumulation, lipid peroxidation, and iron accumulation (Xie et al., 2016). Of note, ferroptosis is proposed to contribute to the lysosomal cell death process (Ye et al., 2021) and has been reported to be negatively regulated by TFEB (Li et al., 2019b). Although it is unknown whether APEX1 is directly involved in ferroptosis, it has been implicated in oxidative stress–induced diseases such as Parkinson's disease, ischemic stroke, Alzheimer's disease, and cancer. Notably, APEX1 overexpression suppresses ROS accumulation, attenuates decreases in the amount of reduced glutathione, and promotes tumor cell survival (Guo et al., 2022; Hao et al., 2019), suggesting that APEX1 may prevent ferroptosis through TFEB *via* Mode I regulation under lysosomal damage. Since our analysis revealed that APEX1 maintains TFEB stability, it will be worth examining possible connections to previous reports that TFEB stability is regulated by STUB1 or THOC4 (Fujita et al., 2021; Sha et al., 2017). How roles of APEX1 in regulating TFEB stability affect TFEB nuclear localization depending on stimuli is currently unclear. But it could be possible that preferential association of APEX1 with TFEB during specific stimuli including lysosomal damage is essential for the maintenance of the TFEB level in the nucleus and ultimately affects TFEB nuclear translocation. Indeed, APEX1 did not show up as a TFEB-interacting candidate upon MK6-83 in WT cells. Further studies will be required to corroborate this possibility.

In contrast to Mode I, our RNA-seq analysis suggested that in Mode II, TFEB regulates factors involved in intracellular trafficking, such as cytoplasmic dynein 1 heavy chain 1 (DYNC1H1), and intracellular transport, including inositol 1,4,5-trisphosphate receptor type 2; such factors are involved in contact site formation and calcium transfer between different organelles (Ziegler et al., 2021). Recent evidence suggests that the transfer of materials through contact sites on lysosomes and other organelles such as the ER is critical for lysosomal homeostasis upon injury (Radulovic et al., 2022; Tan and Finkel, 2022). Thus, it will be fascinating to investigate whether the aforementioned TFEB-regulated factors are involved in modulating lysosomal repair through the formation of contact sites. It is unclear why the absence of lipidated ATG8 failed to activate TFEB during Mode II, but our findings suggest that inhibition of TFEB activation is partly mediated by CCT7, TRIP6, or both through their interaction with TFEB. Further

analysis identifying the interacting domains and localizations of these proteins in relation to lipidated ATG8 can clarify the detailed regulatory mechanisms involved.

Overall, the results of our RNA-seq and interactome analyses might reflect the presence of mode-dependent TFEB regulation and downstream, which may be essential for the maintenance of lysosomal homeostasis upon damage; Mode I prevents acute stress–induced cell death, while Mode II promotes lysosomal recovery. Further studies will expand on our understanding regarding TFEB-dependent lysosomal homeostasis.

TFEB is activated not only by lysosomal damage and starvation, but also, as indicated in previous studies, by cellular stressors such as mitochondrial stress (Goodwin et al., 2021; Nezich et al., 2015), DNA damage (Brady et al., 2018a), proteasome inhibition (Li et al., 2019a), and oxidative stress (Martina et al., 2021). Although there are minimal data on the molecular mechanisms driving TFEB nuclear translocation, the unified mechanism of cellular stress–induced TFEB regulation remains elusive. We discovered that TFEB activation under these cellular stressors can be classified relatively simply as being related to either APEX1-mediated Mode I or TRIP6/CCT7-mediated Mode II. Thus, our study provides novel insights and serves as a first step toward a unified understanding of TFEB regulation.

# Materials and methods

## Cell culture and transfection
HeLa Kyoto and Plat-E cells were cultured in DMEM (043-30085; Wako) containing 10% fetal bovine serum and penicillin–streptomycin at 37°C and 5% $CO_2$. HeLa Kyoto cells were provided by S. Narumiya (Kyoto University, Kyoto, Japan). Plat-E cells were provided by T. Kitamura (The University of Tokyo, Bunkyō, Japan). The cells were cultured in EBSS (E2888; Sigma-Aldrich) for 4 h to induce nutrient starvation. Lipofectamine 2000 (Invitrogen) was used for transient transfection, and cells were used in experiments 24 h after transfection.

## siRNA knockdown
20 nM siRNA was transfected into cells using Lipofectamine RNAiMAX (13778150; Invitrogen), and the transfected cells were used in subsequent experiments 48 h later. The siRNAs used in this study are summarized in Table S3. For RagC/D knockdown, three siRNA against RagC and one siRNA for RagD were mixed as shown in the previous report (Napolitano et al., 2020).

## Antibodies and reagents

For western blotting and immunofluorescence, the following primary antibodies were used in human cells in this study: anti-TFEB (rabbit, 1/1,000, 4240; Cell Signaling Technology), anti-phospho-Ser211 TFEB (rabbit, 1/1,000, 37681; Cell Signaling Technology), anti-S6K (rabbit, 1/1,000, 9202; Cell Signaling Technology), anti-phospho-S6K (rabbit, 1/1,000, 9205; Cell Signaling Technology, 9205), anti-GAPDH (rabbit, 1/50,000, 2118; Cell Signaling Technology), anti-α-tubulin (rabbit, 1/1,0000, PM054; MBL), anti-APEX1 (rabbit, 1/4,000, 10203-1-AP; Proteintech), anti-GFP (rabbit, 1/1,000, 598; MBL), anti-Lamp1 (mouse, 1/1,000, sc-20011; Santa Cruz Biotechnology), and anti-FLAG (mouse, 1/1,000, F1804; Sigma-Aldrich). The following secondary antibodies were used for western blotting: horseradish peroxidase (HRP)–conjugated goat anti-rabbit IgG (1/10,000, 111-035-003; Jackson ImmunoResearch) and HRP-conjugated goat anti-mouse IgG (1/10,000, 115-035-003; Jackson ImmunoResearch). The following secondary antibodies were used for immunofluorescence: goat anti-rabbit IgG H&L (Alexa Fluor 488) (1/1,000, ab150085; Abcam), goat anti-rabbit IgG H&L (Alexa Fluor 568) (1/1,000, ab175695; Abcam), goat anti-mouse IgG H&L (Alexa Fluor 488) (1/1,000, ab150117; Abcam), and goat anti-mouse IgG H&L (Alexa Fluor 568) (1/1,000, A11004; Invitrogen). The following reagents were used in this study: LLOMe (1 mM, L7393; Sigma-Aldrich), MK6-83 (100 μM, 5547; Tocris), oligomycin (10 μM, 495455; Millipore), antimycin A (4 μM, A8674; Sigma-Aldrich), valinomycin (10 ng/ml, V0627; Sigma-Aldrich), etoposide (100 μM, 2200; Cell Signaling Technology), sodium arsenite (100 μM, sc-250986; Santa Cruz Biotechnology), MG132 (15 μM, 474790; Millipore), and chloroquine (100 μM, C6628; Sigma-Aldrich).

## Plasmid construction

Isolation of WT HeLa cell total RNA and cDNA synthesis were performed using RNeasy Plus Mini Kit (74134; QIAGEN) and SuperScript IV Reverse Transcriptase (18090050; Thermo Fisher Scientific). Human APEX1, CCT7, TRIP6, and CNBP cDNA fragments were amplified using the primers shown below, and these fragments were then cloned into the BamHl/NotI site of the pENTR vector.

pMRX-IRES-puro and pMRX-IRES-bsr vectors were kindly provided by Dr. S. Yamaoka (Institute of Science Tokyo, Bunkyō, Japan). pMRX constructs were generated to encode Myc::Parkin (#23955; Addgene), HA::Parkin, mNG::APEX1, mNG::APEX1 mutants resistant to APEX1 KD, carrying a deletion of nuclear localization signal (APEX1 Δ1-33) and a point mutation (APEX1 C65A), TFEB::mNG, and TFEB::EGFP using the LR reaction (11791043; Invitrogen). For the preparation of retroviruses, Plat-E cells were transfected with the pMRX plasmid together with pCMV-VSV-G using PEI and the virus was collected from the supernatant (Saitoh et al., 2003). Cells were infected with retrovirus, and stable expressed transformants were selected with 1 μg/ml puromycin or 5 μg/ml blasticidin. pcDNA3.1 was purchased from Invitrogen (V79020). pcDNA constructs were generated to encode 3×FLAG::APEX1, 3×FLAG::CCT7, mNG::CCT7, 3×FLAG::TRIP6, mNG::TRIP6, 3×FLAG::CNBP, and mNG::CNBP

using the LR reaction following the manufacturer's protocol (11791043; Invitrogen):

APEX1_Fw, 5′-CTTTGAATTCGGATCCATGCCGAAGCGTGGGAAAAAG-3′;
APEX1_Rv, 5′-GTACCGCATGCGGCCGCTCACAGTGCTAGGTATAGGGTGATAGG-3′;
CCT7_Fw, 5′-CTTTGAATTCGGATCCATGATGCCCACACCAGTTATCCTATTGAAAG-3′;
CCT7_Rv, 5′-GTACCGCATGCGGCCGCTCAGTGGGGGCGGCCACG-3′;
TRIP6_Fw, 5′-CTTTGAATTCGGATCCATGTCGGGGCCCACCTGG-3′;
TRIP6_Rv, 5′-GTACCGCATGCGGCCGCTCAGCAGTCAGTGGTGACGG-3′;
CNBP_Fw, 5′-CTTTGAATTCGGATCCATGAGCAGCAATGAGTGCTTCAAG-3′;
CNBP_Rv, 5′-GTACCGCATGCGGCCGCTTAGGCTGTAGCCTCAATTGTGC-3′.

## Immunoprecipitation and MS analysis

Cells stably expressing TFEB::mNG proteins were grown in a 10-cm-diameter dish and treated with or without LLOMe for 1 or 3 h, or with MK6-83 for 3 h, then cross-linked with 0.1% formaldehyde (063-04815; Wako) for 10 min at room temperature. After cross-linking was quenched with 100 mM glycine for 4 min at room temperature, cells were washed with HEPES saline containing 20 mM HEPES-NaOH buffer (pH 7.5) and 137 mM NaCl, and lysed on ice in HEPES-RIPA buffer containing 20 mM HEPES-NaOH buffer (pH 7.5), 150 mM NaCl, 1 mM EGTA, 1 mM $MgCl_2$, 0.25% (wt/vol) Na-deoxycholate, 0.05% SDS, 1% (vol/vol) NP-40, 0.2% (vol/vol) Benzonase nuclease (70746; Millipore), PhosSTOP (4906837001; Roche), and Protease Inhibitor Cocktail (11873580001; Merck). After sonication and centrifugation at 20,380 × g at 4°C for 15 min, the supernatants were incubated with mNG-Trap magnetic agarose beads (ntma-20; ChromoTek) at 4°C for 2 h. The beads were washed four times with HEPES-RIPA buffer and then twice with 50 mM ammonium bicarbonate. Proteins on the beads were digested by adding 200 ng trypsin/Lys-C mix (Promega) at 37°C overnight. The resultant digests were reduced, alkylated, acidified, and desalted using GL-Tip SDB. The eluates were evaporated and dissolved in 0.1% trifluoroacetic acid and 3% acetonitrile (ACN). Liquid chromatography-MS/MS analysis of the resultant peptides was performed on an EASY-nLC 1200 UHPLC connected to an Orbitrap Fusion mass spectrometer through a nanoelectrospray ion source (Thermo Fisher Scientific). The peptides were separated on a C18 reversed-phase column (75 μm × 150 mm; Nikkyo Technos) with a linear 4–32% ACN gradient for 0–100 min, followed by an increase to 80% ACN for 10 min and a final hold at 80% ACN for 10 min. The mass spectrometer was operated in data-dependent acquisition mode with a maximum duty cycle of 3 s. MS1 spectra were measured with a resolution of 120,000, an automatic gain control (AGC) target of 4e5, and a mass range of 375–1,500 m/z. HCD MS/MS spectra were acquired in the linear ion trap with an AGC target of 1e4, an isolation window of 1.6 m/z, a maximum injection time of 35 ms, and a normalized collision

energy of 30. Dynamic exclusion was set to 20 s. Raw data were directly analyzed against the SwissProt database restricted to Homo sapiens using Proteome Discoverer 2.5 (Thermo Fisher Scientific) with the Sequest HT search engine. The search parameters were as follows: (1) trypsin as an enzyme with up to two missed cleavages; (2) precursor mass tolerance of 10 ppm; (3) fragment mass tolerance of 0.6 Da; (4) carbamidomethylation of cysteine as a fixed modification; and (5) acetylation of the protein N terminus and oxidation of methionine as variable modifications.

Peptides were filtered at a false discovery rate of 1% using the Percolator node. Label-free quantification was performed on the basis of the intensities of precursor ions using the Precursor Ions Quantifier node. Normalization was performed such that the total sum of abundance values for each sample over all peptides was the same.

### Immunoprecipitation
Cells stably expressing TFEB::mNG protein were grown in a 10-cm-diameter dish and cultured for 1 or 3 h in the presence of LLOMe, then cross-linked with 0.1% formaldehyde (063-04815; Wako) for 10 min at room temperature. After cross-linking was quenched with 100 mM glycine for 4 min at room temperature, cells were washed with HEPES buffer (20 mM HEPES-NaOH [pH 7.5], 137 mM NaCl) and lysed in ice-cold HEPES-RIPA buffer (20 mM HEPES-NaOH [pH 7.5], 150 mM NaCl, 1 mM EGTA, 1 mM $MgCl_2$, 0.25% [wt/vol] Na-deoxycholate, 0.05% SDS, 1% [vol/vol] NP-40, 0.2% [vol/vol] Benzonase nuclease [70746; Millipore], PhosSTOP [4906837001; Roche], and Protease Inhibitor Cocktail [11873580001; Merck]). After sonication and centrifugation at 20,380 × $g$ at 4°C for 15 min, the supernatants were incubated with mNG-Trap magnetic agarose beads (ntma-20; ChromoTek) at 4°C overnight under gentle rotation. The beads were collected and washed three times with HEPES-RIPA buffer. Immunoprecipitants were eluted from the beads by boiling with 2× sample buffer at 95°C for 10 min.

### Western blotting
Cells were lysed in sample buffer containing 56 mM Tris-HCl (pH 6.8), 6% (vol/vol) glycerol, 2% SDS, 0.1 M DTT, and 2.4% bromophenol blue, and the lysates were boiled at 90°C for 10 min. Samples were subjected to SDS–PAGE and transferred to polyvinylidene fluoride membranes. The membranes were blocked with TBST containing 1% skim milk and incubated at 4°C overnight with primary antibodies diluted in blocking solution. The membranes were washed three times with TBST, incubated for 1 h at room temperature with 5,000× dilutions of HRP-conjugated secondary antibodies in blocking solution, and washed four times with TBST. The immunoreactive bands were detected using Luminate Forte (Millipore) with ChemiDoc Touch Imaging System (Bio-Rad). The intensity of TFEB bands was analyzed using Fiji (version 2.0.0-rc-69/1.52p).

### Immunofluorescence and microscopy
Cells were cultured on coverslips, fixed with 4% paraformaldehyde for 20 min, washed twice with PBS, permeabilized with 50 μg/ml digitonin or 0.1% Triton X-100 in PBS for 10 min, blocked

with 0.2% gelatin in PBS for 30 min, and then incubated with the indicated primary antibodies for 1 h and secondary antibodies for 40 min. After three washes, the coverslips were mounted on slide glass in VECTASHIELD Mounting Medium with DAPI, and the samples were observed with a CQ1 (Yokogawa) using a 40× 0.95 NA dry objective or with a FV3000 confocal microscope (Olympus) operated using FV31S-SW (version 2.3.1.163), or with an IX83 widefield fluorescence microscope (Olympus), using a 60× 1.3 NA oil immersion objective. For the screening, the rate of TFEB nuclear translocation per cell was determined using CQ1 software (Yokogawa, version 1.05.01.01). For the mitochondrial stress, cells with a clear nuclear transition of TFEB were quantified. For the other cellular stresses, the ratio of nuclear to cytoplasmic TFEB::mNG was determined after extraction of nuclear and cytoplasmic contents using CellProfiler (version 3.1.9). The colocalization rate of CCT7 or TRIP6 with Lamp1, and the number of Lamp1 puncta were analyzed using Fiji.

### Time-lapse imaging
Cells were cultured on 3.5-cm-diameter glass-bottom dishes, incubated with reagents, and observed by Dragonfly 200 High-speed Confocal Imaging Platform (Oxford Instruments Andor) equipped with cCMOS camera, Zyla 4.2 Plus (ZYLA-4.2P-USB3; Oxford Instruments Andor) operated by FUSION (version 2.3.0.50), using a 60× 1.42 NA oil immersion objective. One image was taken every 5 min for 5 h. The time-lapse movie was made using Imaris 9.3.1.

### Proximity ligation assay
The PLA was performed using Duolink *In Situ* Detection Reagent Red. Anti-FLAG (mouse, BioLegend) and anti-GFP (rabbit, BioLegend) antibodies were used as primary antibodies. After all steps, the cells were fixed with 4% paraformaldehyde for 20 min, washed twice with PBS, permeabilized with 0.1% Triton X-100 in PBS for 10 min, blocked with 0.2% gelatin in PBS for 30 min, and then incubated with the indicated primary antibodies for 1 h and secondary antibodies for 40 min. After three washes, the coverslips were finally mounted in VECTASHIELD Mounting Medium with DAPI.

### Isolation of RNA and qPCR
Total RNA isolation and cDNA synthesis were performed using RNeasy Plus Mini Kit (74134; QIAGEN) and iScript cDNA Synthesis Kit (1708891; Bio-Rad), respectively. qPCR was performed using Power SYBR Green PCR Master Mix (4367659; Thermo Fisher Scientific). Targets were measured using QuantStudio Real-Time PCR Software version 1.3 (Thermo Fisher Scientific). Actin was used as an internal control.

The primer pair sequences were as follows:

Actin Fw, 5′-ATTGCCGACAGGATGCAGAA-3′;
Actin Rv, 5′-ACATCTGCTGGAAGGTGGACAG-3′;
TFEB Fw, 5′-CGCATCAAGGAGTTGGGAAT-3′;
TFEB Rv, 5′-CTCCAGGCGGCGAGAGT-3′;
TFE3 Fw, 5′-TCCTGAAGGCCTCTGTGGAT-3′;
TFE3 Rv, 5′-AGGTCCAGAAGGGCATCTGA-3′;
MITF Fw, 5′-TGATTCCCAAGTCAAATGATCCA-3′;

MITF Rv 5′-GCAACTTTCGGATATAGTCCACG-3′;
CCL21 Fw, 5′-AGATTCCCGCCAAGGTTGT-3′;
CCL21 Rv, 5′-TGGAGCAGCCTAAGCTTGGT-3′;
IL6 Fw, 5′-AGGGCTCTTCGGCAAATGTA-3′;
IL6 Rv, 5′-GAAGGAATGCCCATTAACAACAA-3′.

## RNA-seq analysis

RNA-seq analysis was performed at the Center of Medical Innovation and Translational Research of Osaka University. The RNA-seq libraries were created using the TruSeq Stranded mRNA Library Prep Kit (Illumina; 20020594), and the sample quality and quantity were determined by capillary electrophoresis. Libraries were examined by qPCR and prepared for immobilization in flow cells using cBot (Illumina), and then sequence-by-synthesis was carried out at Macrogen-Japan using a NovaSeq 6000 S4 system on a NovaSeq 6000.

## Luciferase assay

The luciferase reporter assay was performed with a 2×CLEAR (TFEB RE) reporter of TFEB binding sites (Shin et al., 2016). Briefly, 2×CLEAR (TFEB RE)-luciferase reporters (plasmid #81120; Addgene) and plasmid encoding pMCX-β-galactosidase (gift from Dr. H. Ogawa, The Research foundation for Microbial disease of Osaka University, Suita, Japan) were cotransfected into HeLa cells. After being treated with 1 mM LLOMe for 3 h, cells were lysed at room temperature, and luciferase activity and β-galactosidase activity were measured using a commercially available luciferase assay system (E4030; Promega) and β-galactosidase enzyme assay system (E2000; Promega).

## Statistical analysis and reproducibility

All quantifications are presented as the means ± SD. Statistical analysis was performed using GraphPad 8.0. Experiments with quantitative results were performed three times, and those with nonquantitative results were repeated twice.

## Online supplemental material

Fig. S1 shows regulatory mechanism and downstream targets of TFEB during lysosomal damage. Fig. S2 shows proteomics analysis to identify novel TFEB-interacting proteins during lysosomal damage. Fig. S3 shows APEX1 is identified as a novel Mode I TFEB regulator. Fig. S4 shows APEX1 is required for TFEB expression and stability. Fig. S5 shows CCT7 and TRIP6 are identified as novel Mode II TFEB regulators. Fig. S6 shows knockdown of TRIP6/CCT7 does not affect AMPK signaling. Fig. S7 shows differential requirements of APEX1 or TRIP6/CCT7 for TFEB activation under several cellular stressors. Video 1 shows time-lapse imaging of WT cells expressing TFEB::mNG under LLOMe treatment. Video 2 shows time-lapse imaging of ATG7 KO cells expressing TFEB::mNG under LLOMe treatment. Video 3 shows time-lapse imaging of WT cells expressing TFEB::mNG under MK6-83 treatment. Video 4 shows time-lapse imaging of ATG7 KO cells expressing TFEB::mNG under MK6-83 treatment. Video 5 shows time-lapse imaging of WT cells expressing TFEB::mNG and Myc::Parkin under O/A treatment. Video 6 shows time-lapse imaging of ATG7 KO cells expressing TFEB::mNG and Myc::Parkin under O/A treatment. Table S1 shows a list of DEGs. Table S2 shows GO-term enrichment analysis of Mode I and Mode II TFEB targets. Table S3 shows a list of siRNA used in this study.

## Data availability

All data necessary to evaluate the conclusions in the paper are present in the paper or the online supplemental material. The MS proteomics data have been deposited in the ProteomeXchange Consortium *via* the jPOST partner repository with dataset identifiers PXD043288 and PXD053327. RNA-seq data have been deposited in NCBI GEO with the dataset identifier GSE235340.

## Acknowledgments

We would like to thank K. Takazawa for technical support and Dr. H. Ogawa for providing the pMCX-β-galactosidase plasmids for this research.

Tamotsu Yoshimori was supported by JSPS KAKENHI (22H04982), AMED (JP22gm1410014), and the Takeda Science Foundation. Shuhei Nakamura was supported by AMED (JP24gm1910008), MEXT KAKENHI, a Grant-in-Aid for Transformative Research Areas B (21H05145), JSPS KAKENHI (21H02428, 19K22429, 23K18140, 24H01910, 24K01979), The Uehara Memorial Foundation, Chugai Foundation for Innovative Drug Discovery Science, Toray Science Foundation (23-6408), the Cell Science Research Foundation, the Astellas Foundation for Research on Metabolic Disorders, the Mochida Memorial Foundation for Medical and Pharmaceutical Research, the Mitsubishi Foundation, Research Grants in the Natural Sciences, the NOVARTIS Foundation (Japan) for the Promotion of Science, and Joint Usage and Joint Research Programs of the Institute of Advanced Medical Sciences of Tokushima University, Center for Autophagy and Anti-Aging, Nara Medical University. Shiori Akayama was supported by JSPS KAKENHI (22J11895). Takayuki Shima was supported by JSPS KAKENHI (24K18072). Andrea Ballabio was supported by the Italian Telethon Foundation, the Associazione Italiana per la Ricerca sul Cancro, the Ministero dell'Università e della Ricerca, the European Research Council (AdG; INCANTAR).

Author contributions: S. Akayama: data curation, formal analysis, investigation, methodology, validation, visualization, and writing—original draft, review, and editing. T. Shima: data curation, formal analysis, investigation, methodology, validation, visualization, and writing—original draft, review, and editing. T. Kaminishi: supervision and writing—review and editing. M. Cui: resources and writing—review and editing. J. Monfregola: resources. K. Nishino: investigation, methodology, resources, and validation. A. Ballabio: conceptualization, resources, supervision, and writing—review and editing. H. Kosako: investigation, methodology, resources, and validation. T. Yoshimori: conceptualization, funding acquisition, resources, and supervision. S. Nakamura: conceptualization, funding acquisition, investigation, methodology, project administration, supervision, and writing—original draft, review, and editing.

Disclosures: All authors have completed and submitted the ICMJE Form for Disclosure of Potential Conflicts of Interest. A.

Ballabio reported personal fees from CASMA Therapeutics and personal fees from Avilar Therapeutics outside the submitted work. S. Nakamura reported "I am the cofounder of APGO." No other disclosures were reported.

Submitted: 21 July 2023

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

# Supplemental material

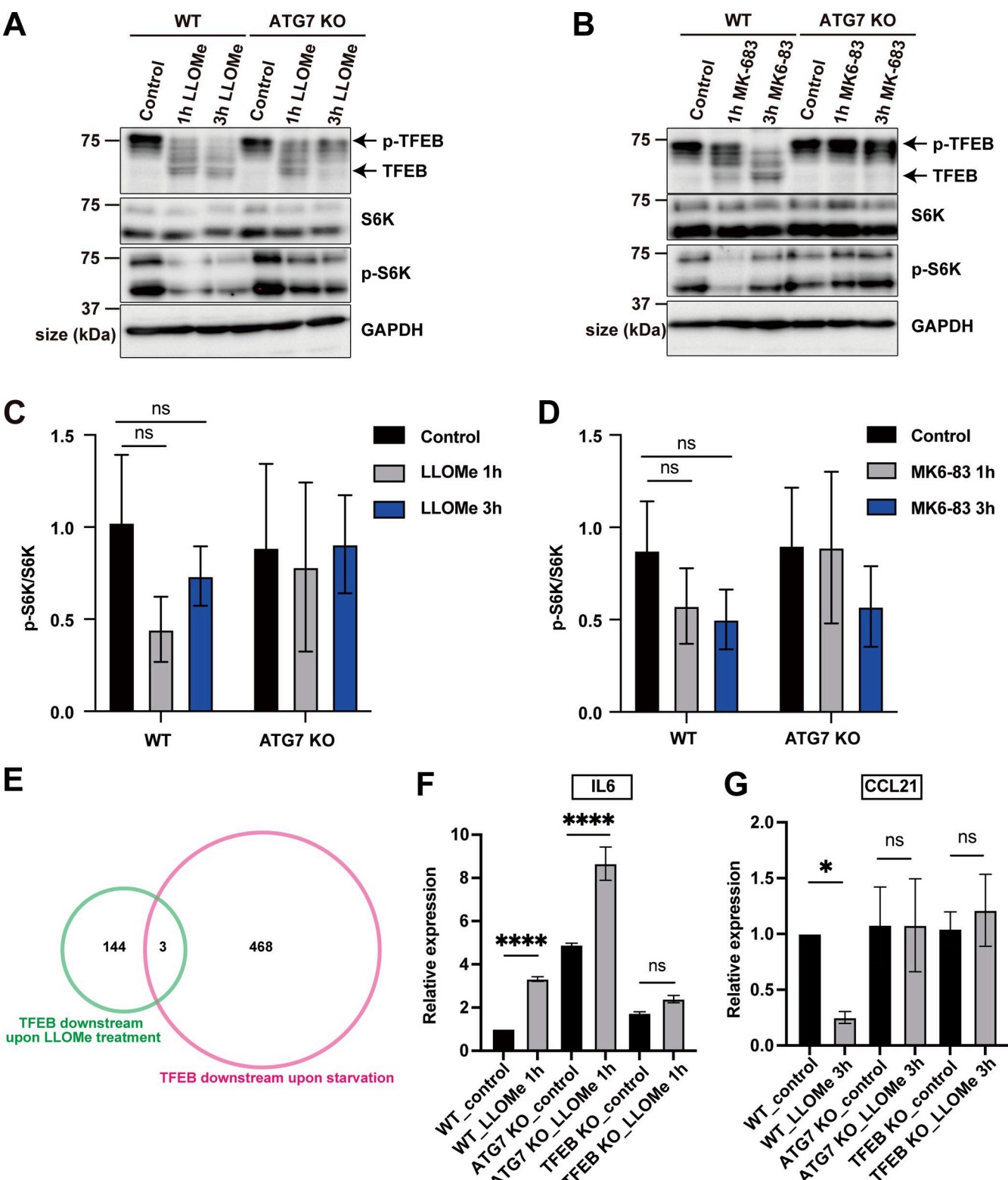

Figure S1. **Regulatory mechanism and downstream targets of TFEB during lysosomal damage. (A and B)** Representative immunoblots of TFEB, S6K, p-S6K, and GAPDH in WT or *ATG7* KO HeLa cells treated with 1 mM LLOMe (A) or 100 µM MK6-83 (B) for the indicated times. **(C and D)** Quantification of image data of pS6K/S6K shown in A and B (*n* = 3 biologically independent samples). **(E)** Venn diagram showing the overlap between two DEGs: 471 genes were TFEB-dependent targets during starvation (pink circle), while 147 genes were TFEB-dependent targets under LLOMe treatment (green circle). Only three genes were regulated under both starvation and LLOMe treatment. **(F and G)** Quantification of the relative expression of IL-6, CCL21 by qPCR in WT, *ATG7* KO, or *TFEB* KO HeLa cells under nontreated(control) or 1 mM LLOMe-treated conditions (*n* = 3 biologically independent samples). **(C, D, F, and G)** P values were determined by ANOVA with Tukey's multiple comparison test; *P < 0.05, ****P < 0.0001. Source data are available for this figure: SourceData FS1.

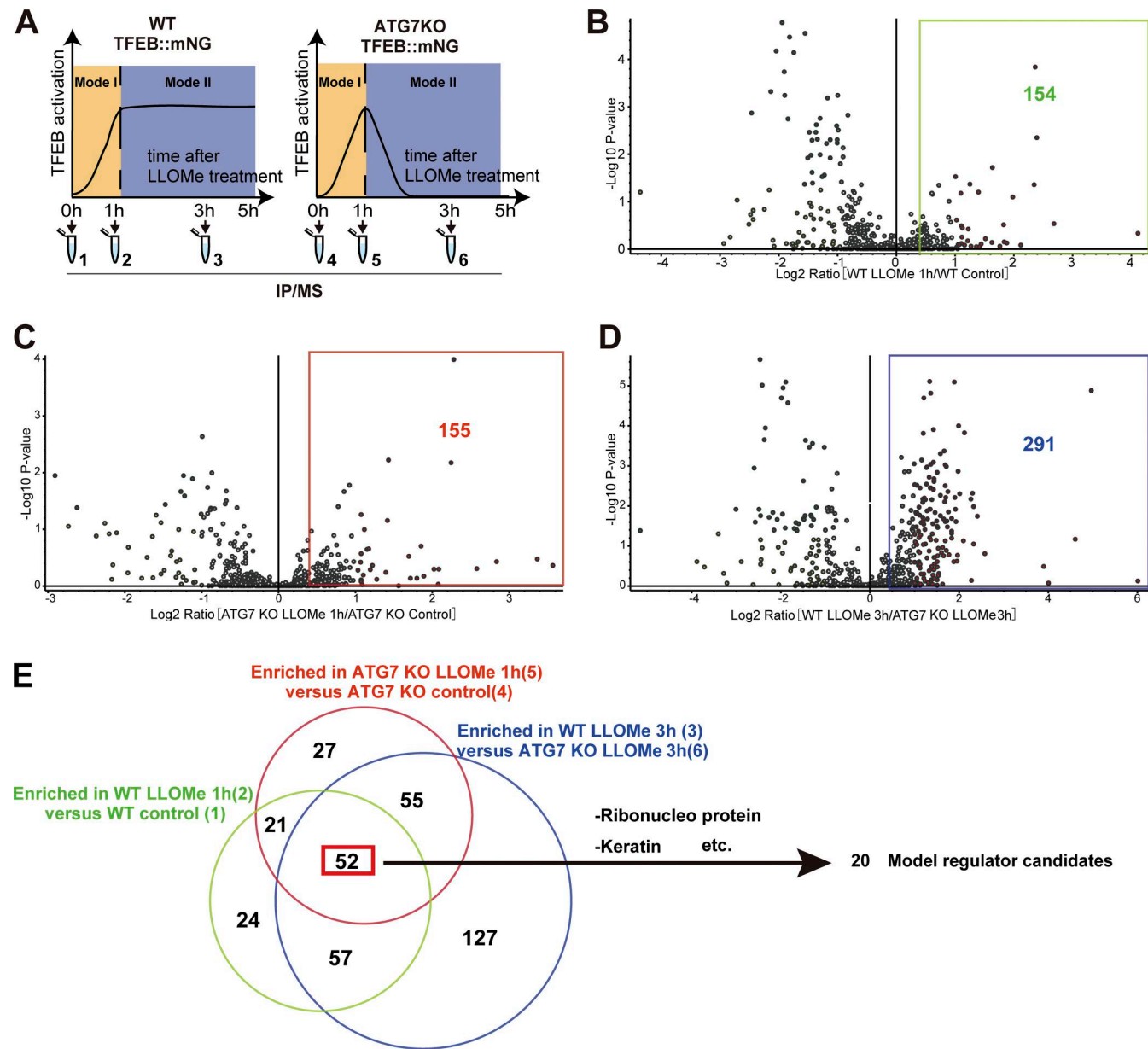

Figure S2. **Proteomics analysis to identify novel TFEB-interacting proteins during lysosomal damage. (A)** Strategy for interactome analysis of TFEB::mNG. Samples of WT and *ATG7* KO HeLa cells stably expressing TFEB::mNG were collected at the indicated time points after 1 mM LLOMe treatment (0 h: untreated control), and subjected to IP followed by MS. **(B)** Volcano plot of the proteins enriched in WT HeLa cells treated with LLOMe for 1 h relative to WT controls (green circle in E) (*n* = 3 biologically independent samples). The green square indicates that the log₂ ratio is >0.4. **(C)** Volcano plot of the proteins enriched in *ATG7* KO HeLa cells treated with LLOMe for 1 h relative to *ATG7* KO controls (red circle in E) (*n* = 3 biologically independent samples). The red square indicates that the log₂ ratio is >0.4. **(D)** Volcano plot of the proteins enriched in WT HeLa cells treated with LLOMe for 3 h relative to *ATG7* KO HeLa cells treated with LLOMe for 3 h (blue circle in E) (*n* = 3 biologically independent samples). The blue square shows that the log₂ ratio is >0.4. **(E)** Venn diagram showing overlap between three sets of proteins in B–D. Fifty-two proteins are enriched under ATG conjugation system–independent TFEB activation. IP, immunoprecipitation.

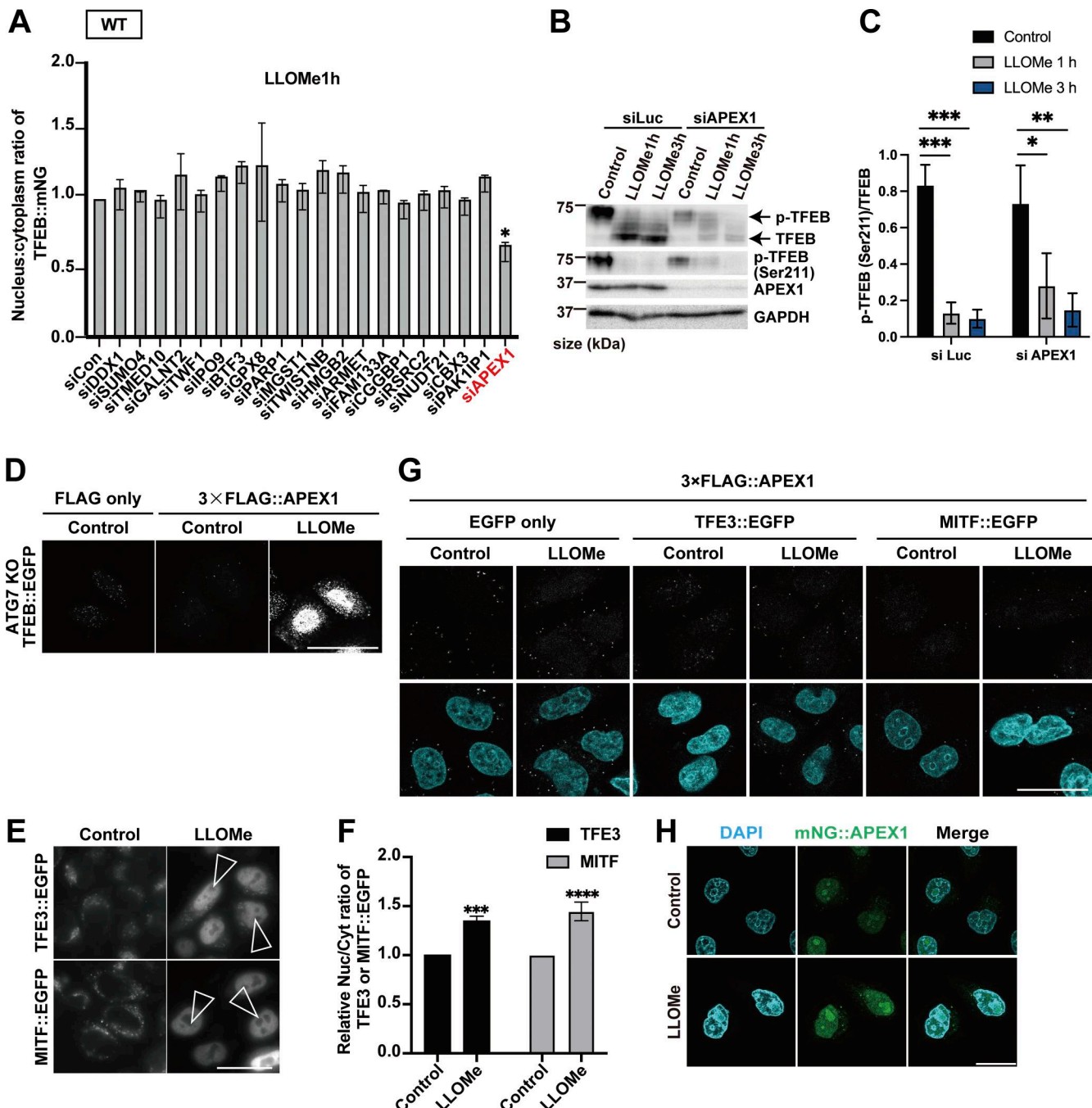

Figure S3.   **APEX1 is identified as a novel Mode I TFEB regulator. (A)** Quantification of TFEB::mNG nuclear-translocated cells in WT HeLa cells stably expressing TFEB::mNG, transfected with scrambled siRNA (siCon) or the indicated siRNA. The cells were treated with 1 mM LLOMe for 1 h ($n = 3$ biologically independent samples). **(B)** Representative immunoblots of TFEB, p-TFEB (Ser211), APEX1, and GAPDH in WT HeLa cells under the indicated conditions. Cells were transfected with siLuc and si*APEX1*. **(C)** Quantification of image data of p-TFEB (Ser211)/TFEB shown in B ($n = 3$ biologically independent samples). **(D)** Results of the PLA using *ATG7* KO HeLa cells stably expressing TFEB::EGFP. The indicated plasmids were transiently transfected and were either untreated (control) or treated with 1 mM LLOMe for 1 h followed by the PLA. **(E)** Representative fluorescence images of WT HeLa cells transiently expressing TFE3::EGFP or MITF::EGFP, either untreated (control) or treated with 1 mM LLOMe for 3 h. **(F)** Quantification of image data shown in E ($n = 3$ biologically independent samples). The relative values of nuclear/cytoplasmic ratios of TFE3 or MITF::EGFP compared with the control are shown. **(G)** Results of the PLA in WT HeLa cells expressing TFE3::EGFP or MITF::EGFP. The indicated plasmids were transiently transfected and were either untreated (control) or treated with 1 mM LLOMe for 3 h followed by the PLA. **(H)** Representative fluorescence images of HeLa cells expressing mNG::APEX1 (green) and either untreated (control) or treated with 1 mM LLOMe for 1 h. **(D, E, G, and H)** Scale bars, 50 μm. **(E)** Arrowheads indicate nuclear-localized TFE3::EGFP or MITF::EGFP. **(C and F)** P values were determined by ANOVA with Tukey's multiple comparison test (C) or *t* test (F); *P < 0.05, **P < 0.01, ***P < 0.001, ****P < 0.0001. Source data are available for this figure: SourceData FS3.

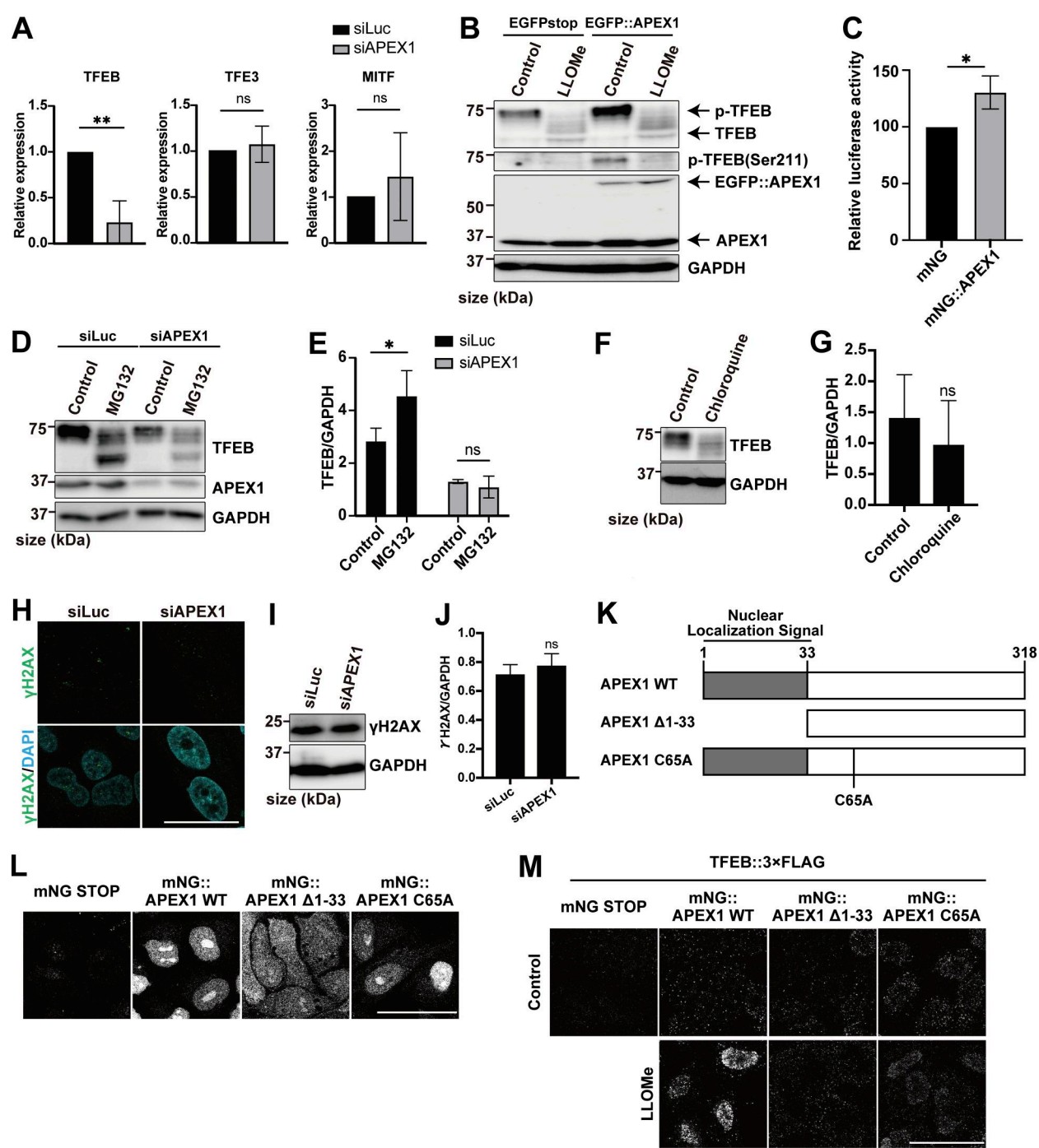

Figure S4. **APEX1 is required for TFEB expression and stability. (A)** Quantification of the relative expression of TFEB, TFE3, and MITF by qPCR in HeLa cells transfected with siLuc or si*APEX1* under nutrient conditions (*n* = 3 biologically independent samples). **(B)** Representative immunoblots of TFEB, p-TFEB (Ser211), APEX1, and GAPDH in WT HeLa cells transfected with EGFP stop (negative control) or EGFP::APEX1, either untreated (control) or treated with 1 mM LLOMe for 1 h. **(C)** Quantification of 2×CLEAR (TFEB RE)-luciferase reporter assays in HeLa cells stably expressing mNG or APEX1::mNG transfected under 1 mM LLOMe treatment for 3-h conditions (*n* = 3 biologically independent samples). **(D)** Representative immunoblots of TFEB, APEX1, and GAPDH in WT HeLa cells that were either untreated (control) or treated with MG132 for 3 h. Cells were transfected with siLuc or si*APEX1*. **(E)** Quantification of the image data shown in D (*n* = 3 biologically independent samples). **(F)** Representative immunoblots of TFEB and GAPDH in WT HeLa cells treated with chloroquine for 4 h. Cells were transfected with si*APEX1*. **(G)** Quantification of image data shown in F (*n* = 3 biologically independent samples). **(H)** Representative immunofluorescence images of HeLa cells stained using an antibody against endogenous γH2AX (green). Cells were transfected with siLuc or si*APEX1*. **(I)** Representative immunoblots of γH2AX and GAPDH in WT HeLa cells that were transfected with siLuc or si*APEX1*. **(J)** Quantification of image data shown in I (*n* = 3 biologically independent samples). **(K)** Schematic representation of APEX1 mutant constructs. **(L)** Representative fluorescence images of HeLa cells stably expressing mNG::APEX1 WT or mNG::APEX1 mutants. **(M)** Results of the PLA using WT HeLa cells stably expressing mNG::stop, mNG::APEX1 WT, or mNG::APEX1 mutants. Cells were transfected with si*APEX1*. The indicated plasmids were transiently transfected and were either untreated (control) or treated with 1 mM LLOMe for 1 h followed by the PLA procedure. **(H, L, and M)** Scale bars, 50 μm. **(E)** P values were determined by ANOVA with Tukey's multiple comparison test; *P < 0.05. **(A and C)** P values were determined by a *t* test; *P < 0.05, **P < 0.01. Source data are available for this figure: SourceData FS4.

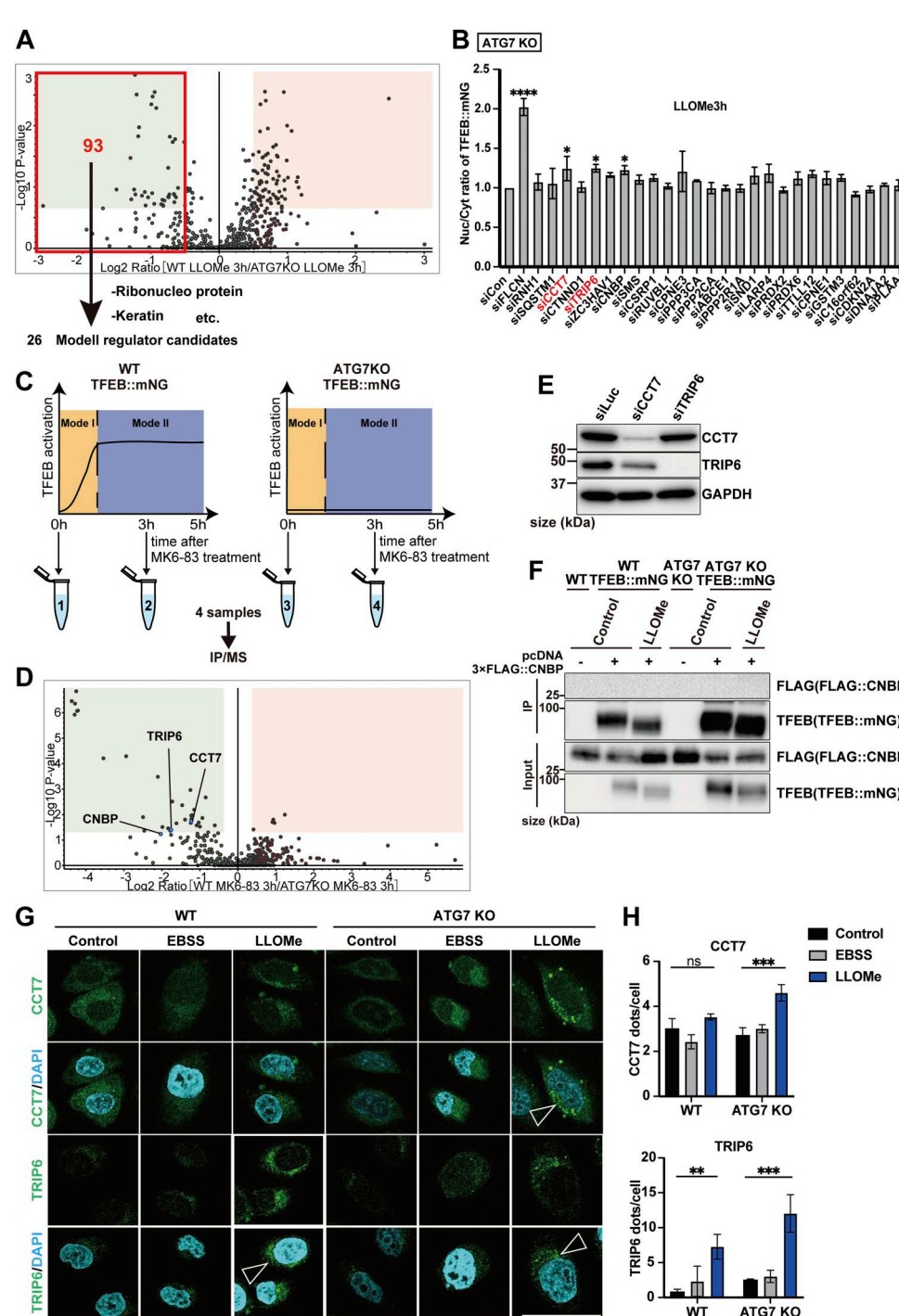

Figure S5. **CCT7 and TRIP6 are identified as novel Mode II TFEB regulators. (A)** Volcano plot of the proteins enriched in WT HeLa cells treated with LLOMe for 3 h relative to *ATG7* KO HeLa cells treated with LLOMe for 3 h (*n* = 3 biologically independent samples). The red square indicates that the log₂ ratio is less than −0.5. 93 proteins are potential Mode II regulators. **(B)** Quantification of TFEB::mNG nuclear-translocated cells in *ATG7* KO HeLa cells stably expressing TFEB:: mNG. Cells were transfected with scrambled siRNA (si Con) or the indicated siRNA and were treated with 1 mM LLOMe for 3 h (*n* = 3 biologically independent samples). **(C)** Strategy for interactome analysis of TFEB::mNG with or without MK6-83 treatment. WT or *ATG7* KO HeLa cells stably expressing TFEB::mNG were collected at the indicated time points after MK6-83 treatment (0 h: untreated control), and subjected to IP/MS. **(D)** Volcano plot of the proteins enriched in WT HeLa cells treated with MK6-83 for 3 h relative to *ATG7* KO HeLa cells treated with MK6-83 for 3 h (*n* = 3 biologically independent samples). **(E)** Representative immunoblots of CCT7, TRIP6, and GAPDH in WT HeLa cells, transfected with siLuc, si*CCT7*, or si*TRIP6* to confirm knockdown efficiency. **(F)** Representative immunoblots of FLAG and TFEB in WT or *ATG7* KO HeLa cells with or without stable TFEB::mNG expression. 3×FLAG::CNBP was transiently transfected, and cells were either untreated (control) or treated with 1 mM LLOMe for 3 h, followed by immunoprecipitation with an antibody against mNG. **(G)** Representative fluorescence images of antibody staining for CCT7 or TRIP6 (green) in WT or *ATG7* KO HeLa cells under either untreated condition (control), starvation, or LLOMe for 3 h. The arrowheads show CCT7/TRIP6 dots. Scale bars, 50 μm. **(H)** Quantification of image data shown in G (*n* = 3 biologically independent samples). **(B and H)** P values were determined by ANOVA with Tukey's multiple comparison test; *P < 0.05, **P < 0.01, ***P < 0.001. Source data are available for this figure: SourceData FS5.

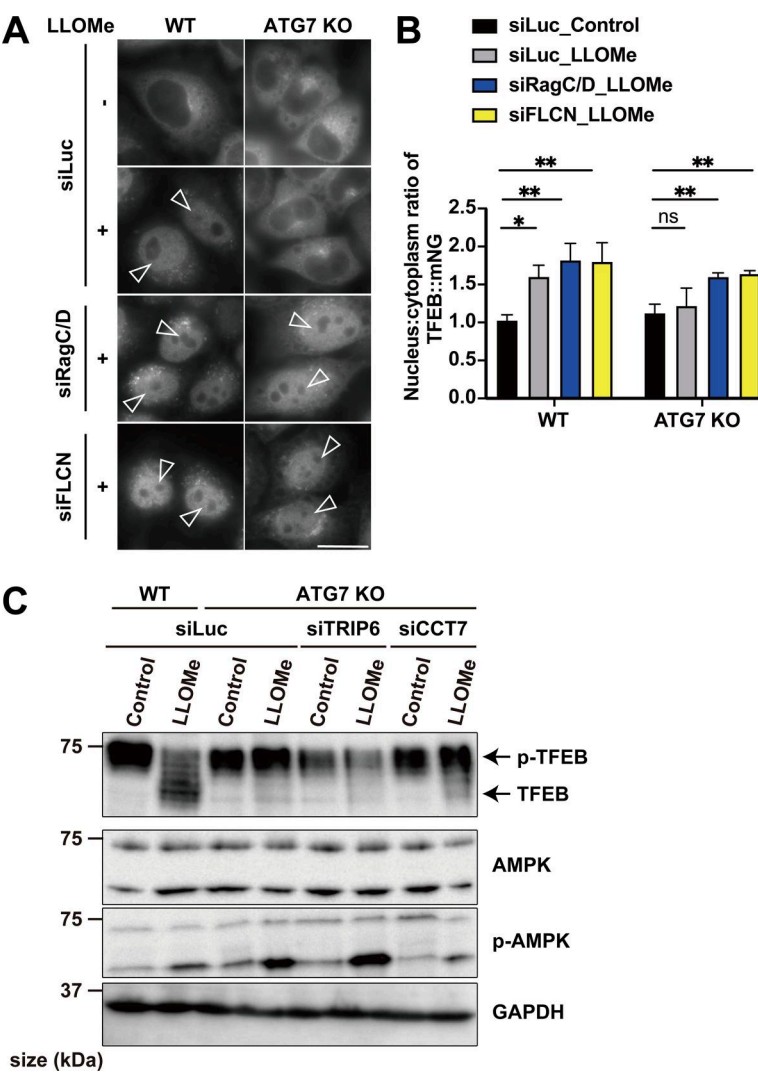

Figure S6. **Knockdown of TRIP6/CCT7 does not affect AMPK signaling. (A)** Representative fluorescence images of WT or *ATG7* KO HeLa cells stably expressing TFEB::mNG, transfected with siLuc, si*RagC/D*, or si*FLCN*. Arrowheads indicate nuclear-localized TFEB::mNG. **(B)** Quantification of image data shown in A (*n* = 3 biologically independent samples). **(C)** Representative immunoblots of TFEB, AMPK, p-AMPK, and GAPDH in WT or *ATG7* KO HeLa cells, transfected with siLuc, si*CCT7*, or si*TRIP6*. **(B)** P values were determined by ANOVA with Tukey's multiple comparison test; *P < 0.05, **P < 0.01. Source data are available for this figure: SourceData FS6.

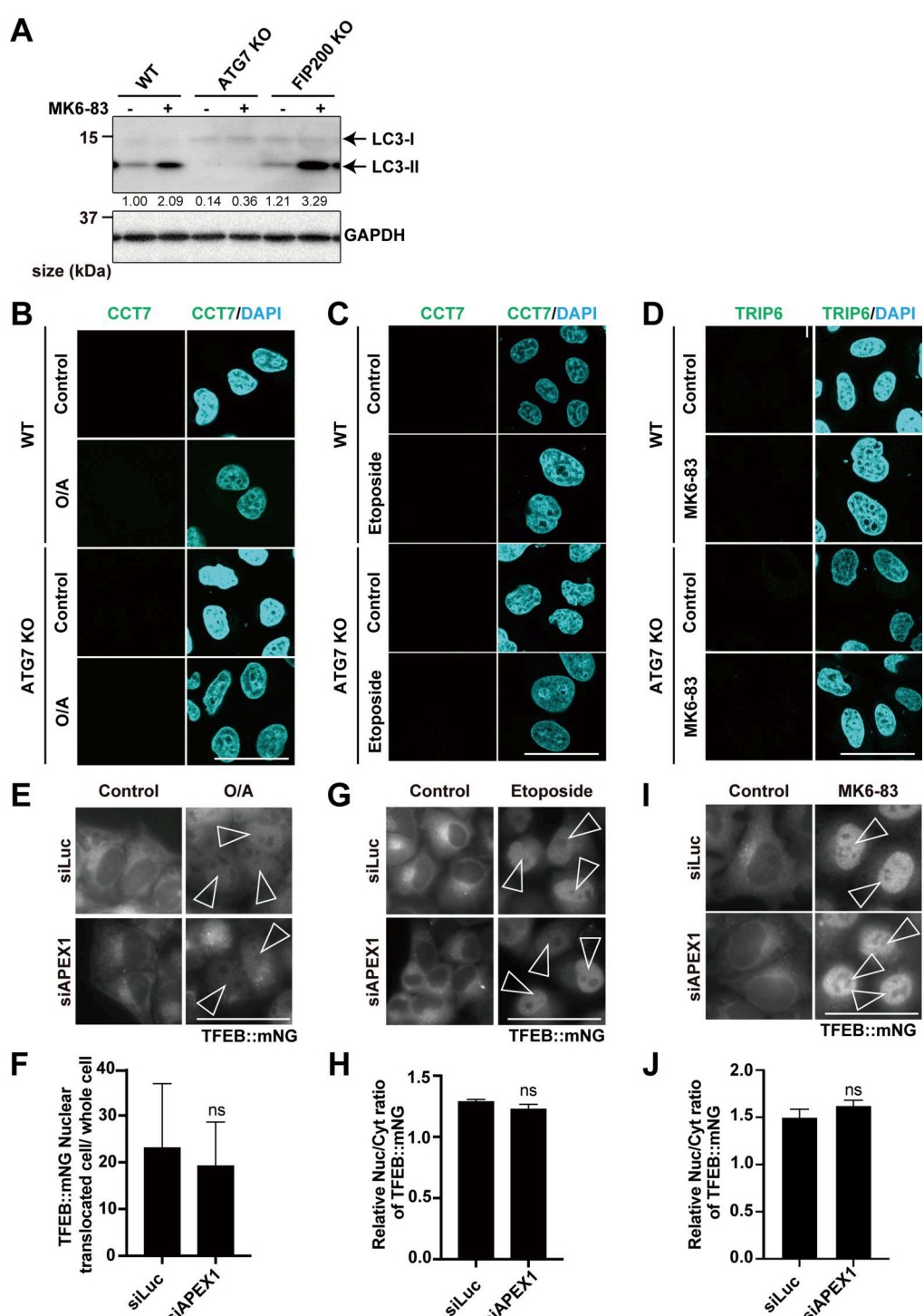

Figure S7. **Differential requirements of APEX1 or TRIP6/CCT7 for TFEB activation under several cellular stressors. (A)** Representative immunoblots of LC3 and GAPDH in WT, *ATG7* KO, or *FIP200* KO HeLa cells were either untreated (control) or treated with MK6-83 for 3 h. Values indicating the amount of LC3-II were normalized to the levels of the GAPDH in each. **(B–D)** Representative fluorescence images of WT or *ATG7* KO HeLa cells transiently expressing mNG::TRIP6 or mNG::CCT7, either untreated (control) or treated with O/A or etoposide or MK6-83. **(E)** Representative fluorescence images of WT HeLa cells stably expressing TFEB::mNG, either untreated (control) or treated with O/A for 3 h. Cells were transfected with siLuc or si*APEX1*. **(F)** Quantification of image data shown in E (*n* = 3 biologically independent samples). **(G)** Representative fluorescence images of WT HeLa cells stably expressing TFEB::mNG, either untreated (control) or treated with etoposide for 24 h. Cells were transfected with siLuc or si*APEX1*. **(H)** Quantification of image data shown in G (*n* = 3 biologically independent samples). **(I)** Representative fluorescence images of WT HeLa cells stably expressing TFEB::mNG, either untreated (control) or treated with MK6-83 for 3 h. Cells were transfected with siLuc or si*APEX1*. **(J)** Quantification of image data shown in I (*n* = 3 biologically independent samples). **(B–D, E, G, and I)** Scale bars, 50 μm. **(E, G, and I)** Arrowheads indicate nuclear-localized TFEB::mNG. **(F, H, and J)** Relative value of the TFEB nuclear translocation rate under drug treatment conditions compared with that under basal (control) conditions. A *t* test was used to compare the mean values of TFEB nuclear translocation rate in each condition. Source data are available for this figure: SourceData FS7.

Video 1.    **Time-lapse imaging of WT cells expressing TFEB::mNG under LLOMe treatment.** Images of TFEB::mNG (green) were captured every 5 min for 5 h using Dragonfly 200 confocal microscopy. A time-lapse movie is created with 10 frames/s.

Video 2.    **Time-lapse imaging of ATG7 KO cells expressing TFEB::mNG under LLOMe treatment.** Images of TFEB::mNG (green) were captured every 5 min for 5 h using Dragonfly 200 confocal microscopy. A time-lapse movie is created with 10 frames/s.

Video 3.    **Time-lapse imaging of WT cells expressing TFEB::mNG under MK6-83 treatment.** Images of TFEB::mNG (green) were captured every 5 min for 5 h using Dragonfly 200 confocal microscopy. A time-lapse movie is created with 10 frames/s.

Video 4.    **Time-lapse imaging of ATG7 KO cells expressing TFEB::mNG under MK6-83 treatment.** Images of TFEB::mNG (green) were captured every 5 min for 5 h using Dragonfly 200 confocal microscopy. A time-lapse movie is created with 10 frames/s.

Video 5.    **Time-lapse imaging of WT cells expressing TFEB::mNG and Myc::Parkin under O/A treatment.** Images of TFEB::mNG (green) were captured every 5 min for 5 h using Dragonfly 200 confocal microscopy. A time-lapse movie is created with 10 frames/s.

Video 6.    **Time-lapse imaging of ATG7 KO cells expressing TFEB::mNG and Myc::Parkin under O/A treatment.** Images of TFEB::mNG (green) were captured every 5 min for 5 h using Dragonfly 200 confocal microscopy. A time-lapse movie is created with 10 frames/s.

**Provided online are Table S1, Table S2, and Table S3. Table S1 shows a list of DEGs. Table S2 shows GO-term enrichment analysis of Mode I and Mode II TFEB targets. Table S3 shows a list of siRNA used in this study.**

