## [Peer Review File · The Journal of Cell Biology]

ATG conjugation dependent/independent mechanisms underlie lysosomal stress induced TFEB regulation

Shiori Akayama, Takayuki Shima, Tatsuya Kaminishi, Mengying Cui, Jlenia Monfregola, Kohei Nishino, Andrea Ballabio, Hidetaka Kosako, Tamotsu Yoshimori, and Shuhei Nakamura

Corresponding Author(s): Shuhei Nakamura, Nara Medical University and Shuhei Nakamura, Nara Medical University

Review Timeline:

Submission Date:	2023-07-21
Editorial Decision:	2023-08-29
Revision Received:	2024-06-27
Editorial Decision:	2024-09-30
Revision Received:	2025-04-30
Editorial Decision:	2025-06-02
Revision Received:	2025-06-16

Monitoring Editor: Hong Zhang

Scientific Editor: Dan Simon

Transaction Report:

DOI: <https://doi.org/10.1083/jcb.202307079>

August 29, 2023

Re: JCB manuscript #202307079

Prof. Shuhei Nakamura
Nara Medical University
Department of Biochemistry
840 Shijo-cho
Kashihara, Nara 6348521
Japan

Dear Prof. Nakamura,

Thank you for submitting your manuscript entitled "ATG conjugation system-dependent/-independent TFEB regulation during lysosomal stress." The manuscript has been evaluated by expert reviewers, whose reports are appended below. Unfortunately, after an assessment of the reviewer feedback, our editorial decision is against publication in JCB.

You will see that Reviewer #2 is supportive and asks for a few additional assays to strengthen the observations and provide controls. However, Reviewers #1&3 both feel that in its present state the work does not provide sufficient insights into the molecular mechanisms by which APEX1, CCT7, and TRIP6 activate TFEB in order to warrant publication in JCB. Additionally, Reviewer #3 requests validation of the mode specific TFEB-interacting proteins in conditions which allow for total separation of the two activation modes.

Given interest in the topic, we would be open to resubmission to JCB of a significantly revised and extended manuscript that fully addresses all of the reviewers' concerns and is subject to further peer-review. If you would like to resubmit this work to JCB, we ask that you please first submit a revision plan detailing the experiments that you will undertake in order to address the reviewer concerns and requests. Please note that priority and novelty will be reassessed at resubmission of the revised manuscript.

Although your manuscript is intriguing, we feel that the points raised by the reviewers are more substantial than can be addressed in a typical revision period. If you wish to expedite publication of the current data, it may be best to pursue publication at another journal.

Regardless of how you choose to proceed, we hope that the comments below will prove constructive as your work progresses. We would be happy to discuss the reviewer comments further once you've had a chance to consider the points raised in this letter. You can contact the journal office with any questions, cellbio@rockefeller.edu or call (212) 327-8588.

Thank you for thinking of JCB as an appropriate place to publish your work.

Sincerely,

Hong Zhang, PhD
Monitoring Editor
Journal of Cell Biology

Dan Simon, PhD
Scientific Editor
Journal of Cell Biology

Reviewer #1 (Comments to the Authors (Required)):

In this manuscript, Akayama S et al. describe ATG conjugation system-dependent and -independent TFEB regulation during lysosomal stress. The authors defined LLOME-induced TFEB nuclear translocation and retention as mode I or II, depending on the requirement for the ATG conjugation system. They then identified different TFEB transcriptomes for these two modes. By analyzing TFEB-interactomes, the authors further identified APEX1 as a mode I TFEB regulator, and CCT7 and TRIP6 as mode II regulator. Interestingly, the authors suggest that TFEB regulation under many different cellular stresses could be categorized into these two modes.

The identification of APEX1, CCT7 and TRIP6 as potential regulators of TFEB is interesting and of help to understand TFEB regulation for lysosome biogenesis/autophagy. Nevertheless, the paper failed to provide mechanistic explanations for how these

factors might regulate TFEB in response to lysosomal damage or other cellular stresses. In addition, how APEX1 and CCT7/TRIP6 are engaged in ATG conjugation system-dependent and -independent TFEB regulation is not explored. In general, this manuscript is premature for publishing in JCB.

Major points:

1. The authors revealed that APEX1 interacts with TFEB in the nucleus. How does the nucleus-localized APEX1 affect TFEB nuclear-cytosol shuttling? Does the interaction of APEX1 with TFEB affect TFEB transcriptional activity? Why does APEX1 specifically affect TFEB expression at the transcription level?
2. There is no evidence that the interaction of APEX1 with TFEB maintains the stability of the latter. What is the real function of APEX1 towards TFEB, i.e., to promote the TFEB gene expression or to stabilize the TFEB protein?
3. Is it possible that loss of APEX1 causes DNA damage, which in turn induces TFEB activation?
4. Does the deficiency of APEX1, CCT7/TRIP6 affect the same genes as described in the mode I or II transcriptomes?
5. What are the explanations for CCT7/TRIP6/CNBP knockdown to rescue the defective nuclear retention of TFEB in ATG7 KO cells induced with LLOME, given that these proteins have different subcellular localizations and should act in different modes.
6. It is interesting that endogenous CCT7 and TRIP6 are evenly distributed in the cytoplasm under normal conditions but translocate to lysosomes following lysosome damage. What are the functions of lysosomal CCT7/TRIP6? How does their interaction with TFEB regulate TFEB activation?
7. It is hard to understand the rationale for classifying potential TFEB regulators into ATG conjugation system-independent (mode I) or -dependent (mode II) groups, because the ATG conjugation system appears to affect TFEB activation indirectly, as previously reported by the authors. In the absence of ATG7, the retention of TFEB in the nucleus (after LLOME treatment) is reduced. What are the possible mechanisms?

Reviewer #2 (Comments to the Authors (Required)):

The transcription factor TFEB is an important regulator of autophagy and lysosome biogenesis that upon various stimuli translocates to the nucleus where it drives its transcriptional activity. TFEB activation and translocation to the nucleus is defective in ATG8 conjugation deficient cells, but by looking at earlier time points, the authors discover an ATG8 independent mode of TFEB activation. The authors go on to define a Mode I and Mode II of TFEB activation, and discover factors that regulate each of the modes including APEX1 for Mode I and CCT7 and TRIP6 for Mode II. While the mechanistic features are not explored in great detail, the discovery of a previously unknown mode of TFEB activation is of great interest to the field. In addition, the identification of factors regulating these modes also represents a very important advance in addition to also assigning the different modes to different cellular stress conditions, including mitophagy and lysosome stress. Overall, this is an excellent and high quality study with very interesting discoveries into the activation of TFEB that will appeal to a broad readership. The comments/suggestions below are aimed at strengthening the authors conclusions.

Major:

1. Figure 1 (and related to Figure 3I): Can the authors show whether MK6-83 treatment induces ATG8 (e.g. LC3B) lipidation?
2. Figure S3B: The effect on TFEB phosphorylation upon siAPEX1 does not appear to be very strong. It would be beneficial to quantitate ($n=3$) so that the lower levels of TFEB in APEX1 knockdown are taken into account by measuring TFEB S211 ratiometrically to the untreated control.
3. To help strengthen the authors conclusions, it would be beneficial to show APEX1 rescue by expressing an RNAi resistant form APEX1 in Figure 2B and 2D-E
4. Is the lysosome localisation of CCT7 and TRIP6 upon LLoMe treatment observed in ATG7 KO also occurring in WT cells?
5. What were the treatment time points for each of the treatments in Figure 4? Does Mode I activation occur during mitophagy? i.e does TFEB translocate to the nucleus at early time points in ATG7 KO as was observed for LLoMe?
6. Figure 4 C-F: While the knockdown of TRIP6 showed a very strong result for mitophagy (Figure 4A-B), some of the effects in 4C-F are quite small. Does combined knockdown of CCT7 and TRIP6 give a stronger effect? This would help clarify if there is some redundancy in the activity of TRIP6 and CCT7. In addition, can the authors show some evidence for the knockdown efficiency of CCT7 and TRIP6?

Minor:

1. Figure 1D: For each GO term it would be helpful to include in the bar graph the number of genes identified within each term. The number could be placed adjacent to each bar (right hand side) representing each GO term.
2. "Next to identify novel compounds" change compounds to factors.

3. The following sentence is a bit confusing: "By contrast, we found that neither arsenite, which induces oxidative stress, nor the proteasome inhibitor MG132 induced TFEB::mNG nuclear translocation in WT cells or autophagy-deficient FIP200 KO and ATG7 KO cells, suggesting that the ATG conjugation system is unnecessary for stress-induced TFEB activation (Fig. 3E-H)."

For clarity, suggested change to:

"By contrast, we found that arsenite, which induces oxidative stress, and the proteasome inhibitor MG132, induced TFEB::mNG nuclear translocation in both WT cells and autophagy-deficient FIP200 KO and ATG7 KO cells, suggesting that the ATG conjugation system is unnecessary for TFEB activation under these stress conditions (Fig. 3E-H)."

4. Figure 4G-J is referred to in the text but it appears that there is no Figure 4J panel?

Reviewer #3 (Comments to the Authors (Required)):

Previous studies have shown that mTORC1 inhibition and loss of RagC/D or FLCN lead to TFEB dephosphorylation and nuclear translocation. Recent findings suggest that lipidation of ATG8 on single membranes sequesters the FLCN complex and thus the ATG conjugation system is critical for TFEB activation. In the paper by Akayama et al. entitled " ATG conjugation system-dependent/-independent TFEB regulation during lysosomal stress ", the authors identified two modes of TFEB activation mediated by distinct mechanisms. Mode I is independent of the ATG conjugation system and can be activated by LLOMe, but not by the TRPML1 agonist MK6-83. Mode II is dependent on the ATG conjugation system and can be activated by both LLOMe and MK6-83. As analyzed by RNAseq, these two modes of TFEB activation appear to be responsible for distinct sets of target genes. To further investigate the key regulatory genes in these two pathways, the authors examined TFEB-interacting proteins by a proteomic approach. In mode I, APEX1 interacts with TFEB in response to LLOMe treatment and independently of the ATG conjugation system. APEX1 stabilizes TFEB protein in the nucleus, thereby regulating TFEB activation. In mode II, CCT7 and TRIP6 interact with TFEB in ATG7 KO cells but not WT cells and sequester TFEB from nuclear translocation. Next, the authors tested various cellular stressors that activate TFEB and classified them as either mode I (requiring APEX1) or mode II (requiring TRIP6/CCT7). Overall, this work provides some new insights into the different regulatory mechanisms of TFEB. However, in its current form, the manuscript has several flaws that need to be addressed.

1. In Fig. 1, the authors have identified two modes of TFEB activation, which introduces a novel and significant concept in this field. However, the current comparison method employed by the authors to identify interacting proteins acting separately in these two modes appears imprecise. The activation curves of the two TFEBs presented in Fig. 2A might potentially mislead readers. Specifically, there is no data demonstrating that model I exhibits a faster response than model II after LLOMe treatment in WT cells, nor is there evidence supporting continuous activation of model I for 3 hours under this condition. Consequently, the comparison depicted in Fig. S2C between proteins enriched in WT cells versus ATG7 KO cells after 3-hour LLOMe treatment seems unreasonable. Therefore, it is necessary to reconsider the strategy for mass spectrometry data analysis. To identify mode I-specific interacting proteins accurately, a comparison should be made among proteins enriched after LLOMe treatment specifically in ATG7 KO cells. Similarly, to detect mode II-specific binding protein, differential protein enrichment following MK6-83 treatment should be examined in WT cells. Complete separation of the two activation modes is important for this research.

2. It is intriguing that APEX1 exclusively mediates TFEB activation in mode I. Given that APEX1 affects the stability of TFEB, it is more likely that both modes are affected. The authors should provide additional data to elucidate why APEX1 specifically regulates TFEB only in mode I.

3. The RNA-seq data presented in Fig. 1 reveals distinct gene sets targeted by the two modes of TFEB activation. However, it remains unclear why there is such a substantial difference in transcriptional regulation and how these pathways are minimally associated with lysosome biogenesis. Furthermore, the paper lacks verification of the up-regulated genes. In addition, why did the authors pay attention to the genes negatively regulated by TFEB, and how to explain the significance of these genes?

4. Previous literature has reported several key genes involved in regulating TFEB, including mTORC1, RAG, FLCN, etc. It would be valuable for the authors to discuss whether the activation of both modes of TFEB relies on these known regulatory genes as this would greatly enhance the importance of their findings.

5. When investigating the mechanism of mode II, the authors mainly focus on the repressor proteins that regulate TFEB, while neglecting to explore the activators. Furthermore, gene enrichment using LLOMe treatment in ATG7 KO cells compared to WT cells may not be an appropriate approach to identify key proteins regulated by mode II triggers (similar problem as question 1). Why do different triggers that depend on mode II have different dependencies on CCT7 and TRIP6? And stating "TRIP6/CCT7-mediated Mode II" could mislead readers into assuming that TRIP6 and CCT7 are required for activating TFEB, while the data actually suggest that they are negative regulators.

Reviewer#1

In this manuscript, Akayama S et al. describe ATG conjugation system-dependent and -independent TFEB regulation during lysosomal stress. The authors defined LLOME-induced TFEB nuclear translocation and retention as mode I or II, depending on the requirement for the ATG conjugation system. They then identified different TFEB transcriptomes for these two modes. By analyzing TFEB-interactomes, the authors further identified APEX1 as a mode I TFEB regulator, and CCT7 and TRIP6 as mode II regulator. Interestingly, the authors suggest that TFEB regulation under many different cellular stresses could be categorized into these two modes.

The identification of APEX1, CCT7 and TRIP6 as potential regulators of TFEB is interesting and of help to understand TFEB regulation for lysosome biogenesis/autophagy. Nevertheless, the paper failed to provide mechanistic explanations for how these factors might regulate TFEB in response to lysosomal damage or other cellular stresses. In addition, how APEX1 and CCT7/TRIP6 are engaged in ATG conjugation system-dependent and -independent TFEB regulation is not explored. In general, this manuscript is premature for publishing in JCB.

First of all, we thank the reviewer for investing tremendous efforts and time to review our paper. Moreover, we sincerely appreciate the reviewer's constructive and valuable comments. All of the comments raised are quite helpful to improve our manuscripts and thus we revised our manuscript as follows. Specific answers to each question are shown below.

Major points:

1. The authors revealed that APEX1 interacts with TFEB in the nucleus. How does the nucleus-localized APEX1 affect TFEB nuclear-cytosol shuttling?

Does the interaction of APEX1 with TFEB affect TFEB transcriptional activity?

Why does APEX1 specifically affect TFEB expression at the transcription level?

We speculated that APEX1 is essential for the stabilization of TFEB after its nuclear translocation. APEX1 is known to regulate the activity of several transcription factors. Since the nuclear localization and/or the redox activity of APEX1 affect its activity, we asked if these domains are required for the interaction with TFEB and the stability of TFEB. We found that neither the nuclear localization signal depleted APEX1 mutant (delta 1-33) nor the redox inactive APEX1 mutant (C65A) interact with TFEB as revealed by PLA assay (Fig. S4N).

Fig.S4

Furthermore, reintroduction of these siRNA resistant APEX1 mutants in an APEX1 knockdown cells did not rescue the expression level of TFEB (Fig.2E, F).

Fig.2

TFEB belongs to MITF/TFE family of transcription factors. To check the specificity of regulation by APEX1 we examined the interaction between APEX1 and other family members, TFE3 and MITF and found that no interaction was observed as revealed PLA assay (Fig. S3G). In addition, although we found lysosomal damage induce the nuclear localization of TFE3 and MITF, knockdown of APEX1 did not affect their nuclear localization (Fig. S3E, F). These results indicate that the interaction and stability of TFEB is at least partly mediated by nuclear localization and redox activity of APEX1 and this regulation is specific to TFEB among MITF/TFE family.

Fig.S3

2. There is no evidence that the interaction of APEX1 with TFE3 maintains the stability of the latter. What is the real function of APEX1 towards TFE3, i.e., to promote the TFE3 gene expression or to stabilize the TFE3 protein?

Thank you. As shown in the response to the previous question. We found that at least nuclear localization and redox activity of APEX1 is essential to its interaction with TFE3 and the TFE3 stability. Moreover, this regulation is specific to TFE3 among MITF/TFE family. We would like to examine how these domains affect TFE3 stability in our future study.

3. Is it possible that loss of APEX1 causes DNA damage, which in turn induces TFE3 activation?

Thank you for the suggestion, To address this question, we examined DNA damage by checking the DSB marker, γ H2AX foci (Fig.S4I-K). At least in our experimental condition, obvious DNA damage was observed by knockdown of APEX1, implying that DNA damage is not involved in TFEB activation during ATG conjugation independent TFEB regulation (Mode I).

Fig.S4

4. Does the deficiency of APEX1, CCT7/TRIP6 affect the same genes as described in the mode I or II transcriptomes?

We validated the expression of IL6 and CCL21 identified from our RNAseq were truly ATG7-independent/TFEB-dependent (Mode I) or ATG7-dependent/TFEB-dependent (Mode II), respectively by qPCR(Fig.S1F, G). However, the impact of APEX1, CCT7/TRIP6 deficiency on these factors were not examined, since we could not determine which time points we should validate the effect of knockdown. In future study, we would like to conduct comprehensive transcriptome analysis to examine how much of mode I and mode II targets are really regulated by APEX1 and CCT7/TRIP6.

Fig. S1

5. What are the explanations for CCT7/TRIP6/CNBP knockdown to rescue the defective nuclear retention of TFEB in ATG7 KO cells induced with LLOME, given that these proteins have different subcellular localizations and should act in different modes.

This is an important question. Our previous report found that the non-canonical function of lipidated LC3 mediated on lysosomes is essential for the activation of TFEB (Nakamura et al., Nat Cell Biol, 2020). LC3 on lysosomes interacts with the lysosomal calcium channel TRPML1, promoting calcium efflux from lysosomes, which leads to the dephosphorylation and activation of TFEB. In the absence of ATG7, lysosomal calcium efflux is impaired, but the underlying mechanism is unclear. CCT7/TRIP6 localizes to lysosomes upon lysosomal damage in ATG7 KO cells, suggesting the possibility that CCT7/TRIP6 binds to TRPML1 in the absence of lipidated LC3 and this inhibits calcium release from lysosomes. However, we could not observe the specific interaction between TRPML1 and CCT7/TRIP6 (Figure1 for the reviewer).

Figure 1 for the reviewer

We also examined the possible cross talk between TRIP6/CCT7 and known regulators of TFEB. mTORC1 activity negatively regulates TFEB activity through the phosphorylation of Ser211 on TFEB. AMPK activation leads to TFEB nuclear localization. However, knockdown of TRIP6/CCT7 in ATG7KO cells neither affects the phosphorylation status of Ser211 on TFEB, nor AMPK (Fig.S6C,D). Although we could not find the exact mechanism by which the knockdown of TRIP6/CCT7 rescues TFEB nuclear localization in ATG7 KO, our it is at least mediated either downstream or parallel of mTORC1 and AMPK signaling.

Fig.S6

6. It is interesting that endogenous CCT7 and TRIP6 are evenly distributed in the cytoplasm under normal conditions but translocate to lysosomes following lysosome damage. What are the functions of lysosomal CCT7/TRIP6? How does their interaction with TFEB regulate TFEB activation?

Thank you. As explained in the response to Question 5, we tested several possibilities during the revision but unfortunately so far, we could not identify exact mechanism how CCT7/TRIP6 on lysosome inhibits TFEB nuclear localization. Nevertheless, we observed depending on the stress which requires ATG conjugation system (Mode II), the localization of TRIP6 and CCT7 presumably on lysosomes differs among these stresses (Fig.4C,F,I). For instance, while TRIP6 dots were increased upon O/A treatment, CCT7 dots were not (Fig.4C, Fig.S7C). Consistently, knockdown of TRIP6 rather than CCT7 rescued O/A induced TFEB activation in ATG7 KO cells (Fig.4A, B). The increment of dots for TRIP6 or CCT7 and their requirement to rescue TFEB activation by its knockdown in ATG7 KO cells were also matched in other stress conditions including Etoposide and MK6-83. Therefore, these data imply that the different dependence of CCT7 or TRIP6 is regulated at the level of their localization. These points were also described in the revised manuscript.

Fig.4

7. It is hard to understand the rationale for classifying potential TFEB regulators into ATG conjugation system-independent (mode I) or -dependent (mode II) groups, because the ATG conjugation system appears to affect TFEB activation indirectly, as previously reported by the authors. In the absence of ATG7, the retention of TFEB in the nucleus (after LLOME treatment) is reduced. What are the possible mechanisms?

To further confirm the presence of two different regulatory mechanism, ATG conjugation system dependent (mode II) and independent mechanism (mode I), we validated RNAseq result by qRT-PCR (Fig.S1F, G). We found that IL6 expression induced by lysosomal damage is abolished by TFEB depletion but not by ATG7 depletion. In contrast, CCL21 is suppressed both in TFEB and ATG7 dependent manner during the lysosomal damage. These results again confirm that the presence of two regulatory mode during lysosomal damage.

Fig.S1

Furthermore, since MK6-83 induces TFEB activation solely dependent on ATG conjugation system (Mode II), we conducted interactome analysis of TFEB and identified CCT7 and TRIP6 as interacting candidates of TFEB (Fig.S5C, D), validating again the existence of mode II regulatory mechanism.

Fig.S5

Our previous report indicates that ATG conjugation system regulates dephosphorylation of TFEB and its nuclear localization through calcium efflux from lysosomes, although we could not determine the actual phosphatases working in this context. Actually, our group is now conducting the large scale screening to identify novel phosphatase regulating phosphorylation status of TFEB under lysosomal damage and we would like to publish this work in our future study. As we described in the response to question 5 and 6, although we could not find the exact mechanism by which the knockdown of TRIP6/CCT7 rescues TFEB nuclear localization in ATG7 KO, our data suggests that it is at least mediated either downstream or parallel of mTORC1 and AMPK signaling.

Reviewer #2

The transcription factor TFEB is an important regulator of autophagy and lysosome biogenesis that upon various stimuli translocates to the nucleus where it drives its transcriptional activity. TFEB activation and translocation to the nucleus is defective in ATG8 conjugation deficient cells, but by looking at earlier time points, the authors discover an ATG8 independent mode of TFEB activation. The authors go on to define a Mode I and Mode II of TFEB activation, and discover factors that regulate each of the modes including APEX1 for Mode I and CCT7 and TRIP6 for Mode II. While the mechanistic features are not explored in great detail, the discovery of a previously unknown mode of TFEB activation is of great interest to the field. In addition, the identification of factors regulating these modes also represents a very important advance in addition to also assigning the different modes to different cellular stress conditions, including mitophagy and lysosome stress. Overall, this is an excellent and high quality study with very interesting discoveries into the activation of TFEB that will appeal to a broad readership. The comments/suggestions below are aimed at strengthening the authors conclusions.

First of all, we thank the reviewer for investing tremendous efforts and time in reviewing our paper. We are grateful for the reviewer's comments on our manuscript that "excellent and high quality study with very interesting discoveries". The reviewer's constructive and valuable comments greatly improved our manuscripts. We revised our manuscript based on reviewer's comments as follows.

Major:

1. Figure 1 (and related to Figure 3I): Can the authors show whether MK6-83 treatment induces ATG8 (e.g. LC3B) lipidation?

Similar to other LC3-dependent TFEB activation-inducing stresses, non-canonical lipidation was also occurring during MK6-83 treatment (Fig.S7A).

Fig.S7

2. Figure S3B: The effect on TFEB phosphorylation upon siAPEX1 does not appear to be very strong. It would be beneficial to quantitate (n=3) so that the lower levels of TFEB in APEX1 knockdown are taken into account by measuring TFEB S211 ratiometrically to the untreated control.

We quantified the data from three independent replicates and found that phosphorylation at the Ser211 on TFEB is decreased both in control and APEX1 KD cells, although the decrease is slightly blunted in APEX1 KD especially after 1hr LLOMe treatment (Fig.S3B, C). Nevertheless, we conclude that the regulation of TFEB activity by APEX1 is largely independent from mTORC1 activity. Thank you.

Fig.S3

3. To help strengthen the authors conclusions, it would be beneficial to show APEX1 rescue by expressing an RNAi resistant form APEX1 in Figure 2B and 2D-E

Thank you! As suggested by the reviewer, we conducted rescue experiments by reintroducing the mNG::APEX1 WT in siAPEX1 treated cells and confirmed that wild-type APEX1 indeed rescued the TFEB protein level comparable to siLuc treated cells (Fig.2E, F), further supporting our conclusion.

Fig.2

4. Is the lysosome localisation of CCT7 and TRIP6 upon LLOMe treatment observed in ATG7 KO also occurring in WT cells?

We have already show localization of CCT7 and TRIP6 in WT cells in the original manuscript, but we will again point this out. TRIP6 but not CCT7 dots were increased in WT cells upon LLOMe treatment and presumably localized on lysosomes (Fig.S5G, H).

Fig.S5

5. What were the treatment time points for each of the treatments in Figure 4? Does Mode I activation occur during mitophagy? i.e does TFEB translocate to the nucleus at early time points in ATG7 KO as was observed for LLoMe?

We are very sorry about the insufficient information. In the revised manuscript, we indicated the treatment time points for each experiment in Fig.4. We did timelapse imaging during mitophagy and confirmed that there is no temporal TFEB nuclear translocation suggesting that Mode I activation does not occur during mitophagy (Videos 5, 6). In addition, to corroborate this conclusion functionally, we also found that knockdown of APEX1 did not affect TFEB nuclear translocation during mitophagy (Fig. 5C,D).

Fig.5

6. Figure 4 C-F: While the knockdown of TRIP6 showed a very strong result for mitophagy (Figure 4A-B), some of the effects in 4C-F are quite small. Does combined knockdown of CCT7 and TRIP6 give a stronger effect? This would help clarify if there is some redundancy in the activity of TRIP6 and CCT7. In addition, can the authors show some evidence for the knockdown efficiency of CCT7 and TRIP6?

Thank you. First, we confirmed through Western blot analysis that both CCT7 and TRIP6 were effectively knocked down (Fig.S5E).

Fig.S5

We found that knockdown of CCT7 specifically rescued TFEB activity induced by MK6-83 treatment (Fig.4G, H), while the knockdown of TRIP6 rescued TFEB activity induced by mitochondrial stress (Fig.4A, B) and DNA damage (Fig.4D, E).

Fig.4

Moreover, additional experiments revealed that CCT7 but not TRIP6 form dot structures during MK6-83 treatment (Fig.4I and Fig.S7D), whereas TRIP6 but not CCT7 form dot structures during mitochondrial stress (Fig.4C and Fig.S7B) and DNA damage (Fig.4F and Fig.S7C)(only main figures are shown below). These results suggest that CCT7 and TRIP6 block TFEB activity through distinct stress-specific pathways and this is regulated at the level of their localization. For these reasons, we did not examine the effect of double knockdown. These points are included in the revised manuscript and we hope this explanation is reasonable.

Fig.4

Minor:

1. Figure 1D: For each GO term it would be helpful to include in the bar graph the number of genes identified within each term. The number could be placed adjacent to each bar

(right hand side) representing each GO term.

Agree. We will include the number of genes identified adjacent to each bar in the revised manuscript.

2. "Next to identify novel compounds" change compounds to factors.

Sorry. We changed.

3. The following sentence is a bit confusing: "By contrast, we found that neither arsenite, which induces oxidative stress, nor the proteasome inhibitor MG132 induced TFEB::mNG nuclear translocation in WT cells or autophagy-deficient FIP200 KO and ATG7 KO cells, suggesting that the ATG conjugation system is unnecessary for stress-induced TFEB activation (Fig. 3E-H)."

For clarity, suggested change to:

"By contrast, we found that arsenite, which induces oxidative stress, and the proteasome inhibitor MG132, induced TFEB::mNG nuclear translocation in both WT cells and autophagy-deficient FIP200 KO and ATG7 KO cells, suggesting that the ATG conjugation system is unnecessary for TFEB activation under these stress conditions (Fig. 3E-H)."

Agree. We changed accordingly. Thank you!

4. Figure 4G-J is referred to in the text but it appears that there is no Figure 4J panel?

We deeply apologize for its error. We removed 4J from the revised manuscript.

Reviewer#3

Previous studies have shown that mTORC1 inhibition and loss of RagC/D or FLCN lead to TFEB dephosphorylation and nuclear translocation. Recent findings suggest that lipidation of ATG8 on single membranes sequesters the FLCN complex and thus the ATG conjugation system is critical for TFEB activation. In the paper by Akayama et al. entitled " ATG conjugation system-dependent/-independent TFEB regulation during lysosomal stress ", the authors identified two modes of TFEB activation mediated by distinct mechanisms. Mode I is independent of the ATG conjugation system and can be activated by LLOMe, but not by the TRPML1 agonist MK6-83. Mode II is dependent on the ATG conjugation system and can be activated by both LLOMe and MK6-83. As analyzed by RNAseq, these two modes of TFEB activation appear to be responsible for distinct sets of target genes. To further investigate the key regulatory genes in these two pathways, the authors examined TFEB-interacting proteins by a proteomic approach. In mode I, APEX1 interacts with TFEB in response to LLOMe treatment and independently of the ATG conjugation system. APEX1 stabilizes TFEB protein in the nucleus, thereby regulating TFEB activation. In mode II, CCT7 and TRIP6 interact with TFEB in ATG7 KO cells but not WT cells and sequester TFEB from nuclear translocation. Next, the authors tested various cellular stressors that activate TFEB and classified them as either mode I (requiring APEX1) or mode II (requiring TRIP6/CCT7). Overall, this work provides some new insights into the different regulatory mechanisms of TFEB. However, in its current form, the manuscript has several flaws that need to be addressed.

We thank the reviewer for suggesting several flaws to become mature. We are grateful for the reviewer's comments on our manuscript that "this work provides some new insights into the different regulatory mechanisms of TFEB". All of the comments raised are quite helpful to improve our manuscripts and thus we revised our manuscript as follows, based on the reviewer's comments.

1. In Fig. 1, the authors have identified two modes of TFEB activation, which introduces a novel and significant concept in this field. However, the current comparison method employed by the authors to identify interacting proteins acting separately in these two modes appears imprecise. The activation curves of the two TFEBs presented in Fig. 2A might potentially mislead readers.

Specifically, there is no data demonstrating that model I exhibits a faster response than model II after LLOMe treatment in WT cells, nor is there evidence supporting continuous activation of model I for 3 hours under this condition.

In ATG7 KO, transient nuclear localization of TFEB was observed(Fig.1A, Video2). However, as pointed out by the reviewer, this does not necessarily imply that Mode I is faster than Mode II. Therefore, we agree that Figure 2A (now Fig.S2A in the revised manuscript) could potentially mislead readers. Instead of highlighting the difference between Mode I and Mode II, we have modified the graph to depict the observed dynamics of TFEB activation over time following LLOMe treatment in both WT and ATG7 KO (Fig.S2A). Moreover, we added the following sentence to admit the complete separation of two modes is difficult but plan to identify novel interacting partners based on the time points based on the time lapse imaging in the result section: ‘Although the complete separation of two regulatory modes is difficult, our time-lapse analysis suggests that Mode I regulation mainly starts within 1h after LLOMe treatment, while Mode II regulation becomes apparent 1h after LLOMe.’

Fig.S2

Although it is a consequence, but to address the reviewer's comment on the persistence of Mode I after LLOMe administration, we investigated whether APEX1, the regulator of Mode I, continues to interact with TFEB after 3h LLOMe. We found that APEX1 continued to bind to TFEB, implying mode I persist even after 3hrs of LLOMe treatment (Fig.2B).

Fig.2

Consequently, the comparison depicted in Fig. S2C between proteins enriched in WT cells versus ATG7 KO cells after 3-hour LLOMe treatment seems unreasonable. Therefore, it is necessary to reconsider the strategy for mass spectrometry data analysis. To identify mode I-specific interacting proteins accurately, a comparison should be made among proteins enriched after LLOMe treatment specifically in ATG7 KO cells. Similarly, to detect mode II-specific binding protein, differential protein enrichment following MK6-83 treatment should be examined in WT cells. Complete separation of the two activation modes is important for this research.

We designate Mode I as ATG conjugation independent regulation. We think that Mode I candidate proteins should be enriched both in WT and ATG7 KO cells at 1h after LLOMe, since TFEB is nuclear localized in both WT and ATG7 KO at this timepoint. On the other hand, at 3h after LLOMe, TFEB is nuclear localized only in WT but not in ATG7 KO. Enriched interacting proteins in ATG7 KO at 3h LLOMe should be present in cytosol to interact with TFEB. Therefore, we still think it is still reasonable to extract proteins which interacts with TFEB more in WT compared to ATG7KO at 3h LLOMe for Mode I candidates(Fig.S2D in the revised manuscript). However, to obtain true candidates during Mode II, as the reviewer suggesting, conducting an interactome analysis of TFEB stimulated by MK6-83 treatment is indeed a great idea. We actually did this proteomics analysis and found that both TRIP6 and CCT7 are identified, similar to lysosomal damage, LLOMe (Fig.S5C, D). Thank you so much for this great suggestion!

Fig.S5

2. It is intriguing that APEX1 exclusively mediates TFEB activation in mode I. Given that APEX1 affects the stability of TFEB, it is more likely that both modes are affected. The authors should provide additional data to elucidate why APEX1 specifically regulates TFEB only in mode I.

It is an indeed important point. We further investigated whether APEX1 functions in situations where Mode II-dependent TFEB regulation is evident, such as under stress induced by O/A, Etoposide, and MK6-83. It was revealed that TFEB activation induced by these stresses was not abolished even upon APEX1 knockdown (Fig.5C-H), suggesting that APEX1 is a specific regulator of mode I.

Fig.5

3. The RNA-seq data presented in Fig. 1 reveals distinct gene sets targeted by the two modes of TFEB activation. However, it remains unclear why there is such a substantial difference in transcriptional regulation and how these pathways are minimally associated with lysosome biogenesis. Furthermore, the paper lacks verification of the up-regulated genes. In addition, why did the authors pay attention to the genes negatively regulated by TFEB, and how to explain the significance of these genes?

It is not yet understood why and how different sets of target genes are controlled by each mode of regulation. However, we believe that this difference itself partially supports the existence of different regulatory modes of TFEB.

As pointed out by the reviewer, while the association of target genes with lysosomal biogenesis was weak, similar context-dependent changes in TFEB targets were observed in other conditions such as lipopolysaccharide treatment, inflammation, and DNA damage (K. L. Carey et al., Cell Rep., 2020; N. Pastore et al., Autophagy, 2016; O. A. Brady et al., Elife, 2018). Additionally, factors identified in the transcriptome were validated to confirm that they are truly ATG7-independent/TFEB-dependent or ATG7-dependent/TFEB-dependent (Fig.S1F,G). This data is now included in the revised manuscript. Actually, we have not specifically paid attention to negatively regulated gene by TFEB in Fig.1C. Since we would like to know the process affected in the absence of TFEB or ATG7, we extracted both up and downregulated gene compared to respective controls.

Fig.S1

4. Previous literature has reported several key genes involved in regulating TFEB, including mTORC1, RAG, FLCN, etc. It would be valuable for the authors to discuss whether the activation of both modes of TFEB relies on these known regulatory genes as this would greatly enhance the importance of their findings.

This is indeed a crucial point, and we appreciate your suggestion. We confirmed whether the knockdown of APEX1 affects the mTORC1-dependent Ser211 phosphorylation of TFEB. We found that similar to control knockdown (siLuc), knockdown of APEX1 also decreased phosphorylation of Ser211 on TFEB which is known direct target of mTORC1 (Fig.S3B, C) by LLOMe, suggesting APEX1 works independently from mTORC1 function.

Fig.S3

Furthermore, as mentioned by the reviewer, a previous study (Goodwin et al., *Sci Adv*, 2021) suggests that non-canonical lipidation of GABARAP sequesters FLCN and disrupts its GAP activity towards RagC/D, resulting in dephosphorylation and activation of TFEB. Consistent with this finding, when RagC or FLCN was knocked down, TFEB translocated to the nucleus in ATG7KO cells in response to lysosomal damage (Fig.S6A, B). Thus, it was revealed that ATG conjugation system-dependent TFEB activation during lysosomal damage involves the FLCN-RagC/D-mTORC1 axis.

Fig.S6

The potential crosstalk between this axis and the function of CCT7/TRIP6 was investigated by checking the impact of TRIP6 and CCT7 knockdown on the mTORC1-dependent phosphorylation of TFEB. We observed knockdown of either TRIP6 or CCT7 in ATG7 KO cells did not largely affect phosphorylation of Ser211 on TFEB, which is comparable to that in control knockdown (siLuc) in ATG7 KO cells. Additionally, although TFEB is reported to be regulated by AMPK, knockdown of either TRIP6 or CCT7 did not affect phosphorylation status of AMPK (Fig.S6C, D). These results indicate knockdown of either TRIP6 or CCT7 rescues TFEB nuclear localization in ATG7 KO cells independently or downstream of mTORC1 and AMPK signaling.

Fig.S6

5. When investigating the mechanism of mode II, the authors mainly focus on the repressor proteins that regulate TFEB, while neglecting to explore the activators. Furthermore, gene enrichment using LLOMe treatment in ATG7 KO cells compared to WT cells may not be an appropriate approach to identify key proteins regulated by mode II triggers (similar problem as question 1). Why do different triggers that depend on mode II have different dependencies on CCT7 and TRIP6?

In the initial idea to do this screening is to identify the factors which block TFEB activation in ATG7 KO cells. Therefore, we focused on repressors. We appreciate the reviewer's suggestion in question1, we also conducted proteomics under MK6-83 treatment which only induces mode II dependent TFEB activation. This analysis identified again CCT7 and TRIP6 validating our approach. The reasons for the differential dependency on CCT7 or TRIP6 by various triggers are currently unclear. However, we observed depending on the stress which requires ATG conjugation system(Mode II), the localization of TRIP6 and CCT7 differs among these stresses (Fig.4C,F,I). For instance, while TRIP6 dots were increased upon O/A treatment, CCT7 dots were not (Fig.4C, Fig.S7B, only main Figures are shown below). Consistently, knockdown of TRIP6 rather than CCT7 is required to rescue O/A induced TFEB activation in ATG7 KO cells (Fig.4A). The increment of dots for TRIP6 or CCT7 and their requirement to rescue TFEB activation by its knockdown in ATG7 KO cells were also matched in other stress conditions including Etoposide and MK6-83. Therefore, these data imply that the different dependence of CCT7 or TRIP6 is regulated at the level of their localization. These points were also described in the revised manuscript.

Fig.4

C

F

I

And stating "TRIP6/CCT7-mediated Mode II" could mislead readers into assuming that TRIP6 and CCT7 are required for activating TFEB, while the data actually suggest that they are negative regulators.

Agree. We deleted the term "TRIP6/CCT7-mediated Mode II" to avoid the misleading readers. Instead, we wrote TRIP6/CCT7 function as negative regulators of TFEB under the absence of ATG conjugation system.

September 30, 2024

Re: JCB manuscript #202307079R-A

Prof. Shuhei Nakamura
Nara Medical University
Department of Biochemistry
840 Shijo-cho
Kashihara, Nara 6348521
Japan

Dear Prof. Nakamura,

Thank you for submitting your revised manuscript entitled "ATG conjugation system-dependent/-independent TFEB regulation during lysosomal stress." Please accept our apologies for the delay in getting the reviews back to you and thank you for your patience. The manuscript has been seen by the original reviewers whose full comments are appended below.

You will see that Reviewer #2 is positive about the work in terms of its suitability for JCB. However, Reviewer #1 states that their concerns about the need for more insights into the mechanisms by which APEX1, TRIP6, and CCT7 regulate the nuclear translocation of TFEB, have not been addressed. And Reviewer #3, while overall more positive, still has some remaining questions about the proposed modes of TFEB regulation. Having considered the reviewer feedback over both review rounds we also agree that a clearer understanding of the direct molecular mechanisms of how APEX1, CCT7, & TRIP6 regulate TFEB, would be necessary for further consideration at JCB.

Our general policy is that papers are considered through only one revision cycle. However, given the interest in this topic, we are open to an additional round of revisions. Please note that a substantial amount of additional data would likely be needed to sufficiently address the remaining concerns of Reviewer #1 and to answer the questions raised by Reviewer #3.

If you choose to resubmit, please include a cover letter addressing the reviewers' comments point by point. Please also highlight all changes in the text of the manuscript.

Regardless of how you choose to proceed, we hope that the comments below will prove constructive as your work progresses. We would be happy to discuss them further once you've had a chance to consider the points raised. You can contact the journal office with any questions at cellbio@rockefeller.edu.

Thank you for thinking of JCB as an appropriate place to publish your work.

Sincerely,

Hong Zhang, PhD
Monitoring Editor
Journal of Cell Biology

Dan Simon, PhD
Scientific Editor
Journal of Cell Biology

Reviewer #1 (Comments to the Authors (Required)):

The reviewer appreciates the authors' revision by performing additional experiments and providing new data. Unfortunately, the current revision failed to provide mechanistic clues on how APEX1, TRIP6 and CCT7 regulate TFEB (nuclear translocation) in response to lysosomal damage, dependently or independently on ATG7. Apparently, a lot of in-depth work is needed to suffice the paper for publication.

Reviewer #2 (Comments to the Authors (Required)):

The authors have convincingly addressed the comments resulting in a strengthened manuscript. I have only one minor comment

remaining that is in relation to the original comment number 2:

2. It is great that the data $n=3$ was quantitated. To clarify on the original comment: can the ratio of phospho TFEB be quantitated relative to total TFEB? There is a very strong reduction in total TFEB levels upon siAPEX1 and therefore the reduction in phospho TFEB may be due to less total TFEB protein. Indeed, in the nice experiment conducted in response to comment 3, the authors show that total TFEB levels can be rescued following siAPEX1 and rescue with si resistant form of APEX1.

Reviewer #3 (Comments to the Authors (Required)):

After the revision, most of the questions has been answered and the quality of the manuscript has been improved. However, there are still some issues that need further discussion:

1. When looking for mode I-specific TFEB binding proteins, the candidate proteins are not necessarily affected by mode II. Therefore, I still don't understand the reason for adding criteria in Figure S2D; I think it is more reasonable to select proteins at the intersection according to Figure S2B and S2C.
2. During mode II activation (such as MK6-83 treatment), does nuclear TFEB bind to APEX1? Given that APEX1 affects TFEB stability regardless of the trigger, it is still a bit confusing why APEX1 only affects TFEB activation in mode I. Or stability is not the key reason why APEX1 affects TFEB nuclear translocation in Mode I.
3. According to the results in Figure S5B, the phenotype of FLCN knockdown on TFEB nuclear localization is significantly stronger than that of CCT7 and TRIP6 knockdown. In addition, the connection between FLCN and the ATG conjugation system has been demonstrated in the previous study (Goodwin et al., 2021), while the relationship between CCT7/TRIP6 and ATG conjugation system is still unclear. Would it be reasonable to attribute mode II directly to TFEB activation regulated by the FLCN-Rag pathway?

Reviewer #1 (Comments to the Authors (Required)):

The reviewer appreciates the authors' revision by performing additional experiments and providing new data. Unfortunately, the current revision failed to provide mechanistic clues on how APEX1, TRIP6 and CCT7 regulate TFEB (nuclear translocation) in response to lysosomal damage, dependently or independently on ATG7. Apparently, a lot of in-depth work is needed to suffice the paper for publication.

Response

Thank you so much for taking time and tremendous effort for reviewing our manuscript. To get further mechanistic insight how APEX1, TRIP6 and CCT7 is involved in TFEB regulation, we conducted the following experiments. For APEX1, we have shown that APEX1 which binds to TFEB in the nucleus is essential for the maintenance of TFEB stability. Since we found that TFEB transcripts are reduced in siAPEX1 treated cells, we speculate APEX1 might be involved in transcriptional activity of TFEB which activates own transcription through the CLEAR element (TFEB direct binding sites). Consistent with this idea, we found that mNeonGreen-APEX1 overexpression significantly increases luciferase activity driven by 2x CLEAR elements under LLOMe(Fig.S4C).

FigS4C

Regarding CCT7/TRIP6, we found that knockdown of CCT7/TRIP6 in ATG7 KO cells reduced phosphorylation on Ser211 of TFEB (Fig.3H, I in revised figures), suggested that the rescue of TFEB nuclear localization by siCCT7/TRIP6 in ATG7 KO is mediated through downregulation of mTORC1 activity.

Fig.3H, I

Then, to further search for the significance of CCT7 and TRIP6 functions in the regulation of TFEB, we monitored TFEB activity after LLOMe wash-off in WT. At 18hrs after LLOMe wash off, we found that TFEB returns to the inactive state as revealed by acquisition of phosphorylation on Ser211 and upshift of TFEB total bands in siControl treated cells. In contrast, TFEB is still activated as revealed by its dephosphorylation on Ser211 and slight downshift of TFEB total bands in siCCT7 or siTRIP6 treated cells(Fig.3J). These data suggest that both CCT7 and TRIP6 function as negative regulators to plays a role to terminate TFEB activation at the certain periods of time after the lysosomal damage.

Fig.3J

Previously, Goodwin et al., 2021, revealed that FLCN-FNIP complex is sequestered by GABARAPs upon TRPML1 agonist treatment, ultimately leading to the inhibition of phosphorylation of TFEB by mTORC1 and the induction of nuclear translocation of TFEB. Therefore, we tried to examine the possible cross talk between CCT7/TRIP6 and FLCN-FNIP mediated mTORC1 regulation (Confidential Figure1). In contrast to TRPML1 agonist treatment, FLCN dots appeared even under the

untreated steady state condition and seemed to be slightly increased upon LLOMe in siLuc treated cells. In ATG7 KO, FLCN dots seemed to be further increased upon LLOMe in siLuc treated cells compared to that in WT. We found that siCCT7 or siTRIP6 decreased the FLCN dots in ATG7 KO cells. These data could imply that CCT7 and TRIP6 are involved in the formation of FLCN dots, but its underlying mechanism and physiological consequence are unclear. Thus, these FLCN data were excluded from the revised manuscripts, and we would like to address this possible crosstalk in our future study.

Confidential Figure 1

We principally agree with the reviewer that it will be important to investigate the much-detailed mechanism by which APEX1, CCT7 and TRIP6 regulate TFEB. However, we respectfully argue that such detailed mechanistic studies are beyond the scope of this first manuscript, in which main objective is to report the presence of ATG conjugation dependent and independent regulation of TFEB during lysosomal damage and the identification of novel factors involved in each regulation. We hope these explanations are reasonable and can be accepted.

Reviewer #2 (Comments to the Authors (Required)):

The authors have convincingly addressed the comments resulting in a strengthened manuscript. I have only one minor comment remaining that is in relation to the original comment number 2:

2. It is great that the data n=3 was quantitated. To clarify on the original comment: can the ratio of phospho TFEB be quantitated relative to total TFEB? There is a very strong reduction in total TFEB levels upon siAPEX1 and therefore the reduction in phospho TFEB may be due to less total TFEB protein. Indeed, in the nice experiment conducted in response to comment 3, the authors show that total TFEB levels can be rescued following siAPEX1 and rescue with si resistant form of APEX1.

Response

Thank you so much for positive comments on our revised manuscript and clarifying the original comment no.2. In the previous revision, we quantified phospho TFEB relative to GAPDH. So we re-quantified phospho TFEB relative to total TFEB. Thanks to the reviewer's suggestion, the dephosphorylation of TFEB upon LLOMe is slightly blunted in APEX1 knockdown cells compared to control knockdown, implying that inhibition of mTORC1 activity is partly affected by APEX1 knockdown during lysosomal damage (Fig.S3C). This implies that TFEB nuclear localization could be regulated by APEX1 through affecting mTORC1. As the reviewer pointed out, total TFEB level is reduced in APEX1 knockdown cells under LLOMe, which is rescued by siRNA resistant TFEB expression. Considering APEX1 binds to TFEB mainly in the nucleus, we speculate APEX1 function is mainly required for the maintenance of TFEB level in the nucleus during lysosomal damage and this sustains its nuclear translocation. How roles of APEX1 regulating TFEB stability affect mTORC1 activity is currently unclear. This point is mentioned in our result section.

Fig.S3C

Reviewer #3 (Comments to the Authors (Required)):

After the revision, most of the questions has been answered and the quality of the manuscript has been improved. However, there are still some issues that need further discussion:

1. When looking for mode I-specific TFEB binding proteins, the candidate proteins are not necessarily affected by mode II. Therefore, I still don't understand the reason for adding criteria in Figure S2D; I think it is more reasonable to select proteins at the intersection according to Figure S2B and S2C.

Response

We apologize lack of sufficient explanation. For mode I candidates, we would like to identify the positive regulator and the interacting partners of TFEB which work during whole period of LLOMe treatment. I understand the intersection of Fig.S2B and S2C might be sufficient considering mode I is not necessarily affected by mode II. We added Fig.S2D criteria based on our time lapse imaging, simply narrowing down candidates which work even after 3hr LLOMe, rather than to pick up mode II independent candidates. Around 3hr after LLOMe, most of TFEB start to go back to cytosol in ATG7 KO cell, while TFEB remains in nucleus in WT. Therefore, we hypothesize that at this timepoint(3hrs), interacting candidates which work as TFEB positive regulators should be enriched more in WT (TFEB mainly in nucleus) compared to ATG7 KO (TFEB mainly in cytosol). We added this further explanation and changed several expressions to avoid the confusion in the revised manuscript.

2. During mode II activation (such as MK6-83 treatment), does nuclear TFEB bind to APEX1? Given that APEX1 affects TFEB stability regardless of the trigger, it is still a bit confusing why APEX1 only affects TFEB activation in mode I. Or stability is not the key reason why APEX1 affects TFEB nuclear translocation in Mode I.

Response

Thank you so much for pointing this out. Based on our TFEB interactome data during MK6-83, TFEB could bind to APEX1 but lesser extent compared to LLOMe (Fig.S5C, Confidential Figure1), when the cut-off for protein enrichment was set to a \log_2 fold change of more than 0.4, the same cut-off under the LLOMe condition. Thus, we speculate this preferential association of APEX1 with TFEB during lysosomal damage is essential for the maintenance of TFEB level in the nucleus and ultimately affect TFEB nuclear translocation. This possibility is stated in the discussion of our revised manuscript.

Confidential Figure1

3. According to the results in Figure S5B, the phenotype of FLCN knockdown on TFEB nuclear localization is significantly stronger than that of CCT7 and TRIP6 knockdown. In addition, the connection between FLCN and the ATG conjugation system has been demonstrated in the previous study (Goodwin et al., 2021), while the relationship between CCT7/TRIP6 and ATG conjugation system is still unclear. Would it be reasonable to attribute mode II directly to TFEB activation regulated by the FLCN-Rag pathway?

Response

Thank you so much for the valuable comments. We tested possible crosstalk between CCT7/TRIP6 and FLCN-Rag-mTORC1 pathway. First, we found that knockdown of CCT7/TRIP6 in ATG7 KO cells reduced phosphorylation on Ser211 of TFEB, suggested that the rescue of TFEB nuclear localization by siCCT7/TRIP6 in ATG7 KO is mediated through downregulation of mTORC1 activity.

Fig.3H, I

Then, to further search for the significance of CCT7 and TRIP6 functions in the regulation of TFEB, we monitored TFEB activity after LLOMe wash-off in WT. At 18hrs after LLOMe wash off, we found that TFEB returns to the inactive state as revealed by acquisition of phosphorylation on Ser211 and upshift of TFEB total bands in siControl treated cells. In contrast, TFEB is still activated as revealed by its dephosphorylation on Ser211 and slight downshift of TFEB total bands in siCCT7 or siTRIP6 treated cells. These data suggest that both CCT7 and TRIP6 function as negative regulators to play a role to terminate TFEB activation at the certain periods of time after the lysosomal damage.

Fig.3J

As the reviewer mentioned, Goodwin et al., 2021, revealed that FLCN-FNIP complex is sequestered by GABARAPs upon TRPML1 agonist treatment, ultimately leading to the inhibition of phosphorylation of TFEB by mTORC1 and the induction of nuclear translocation of TFEB. In contrast to TRPML1 agonist treatment, FLCN dots appeared even under the untreated steady state condition and seemed to be slightly increased upon LLOMe in siLuc treated cells (Confidential Figure 2). In ATG7 KO, FLCN dots seemed to be further increased upon LLOMe in siLuc treated cells compared to that in WT. We found that siCCT7 or siTRIP6 decreased the FLCN dots in ATG7 KO cells. These data could imply that CCT7 and TRIP6 are involved in the formation of FLCN dots, but its underlying mechanism and physiological consequence are unclear. Thus, these FLCN data were excluded from the revised manuscripts and we would like to address this possible cross talk in our future study.

Confidential Figure 2

June 2, 2025

RE: JCB Manuscript #202307079RR

Shuhei Nakamura
Nara Medical University

Dear Prof. Nakamura,

Thank you for submitting your revised manuscript entitled "ATG conjugation system-dependent and -independent mechanisms underlie lysosomal stress-induced TFEB regulation." We would be happy to publish your paper in JCB as a Report pending final revisions necessary to meet our formatting guidelines (see details below).

A. MANUSCRIPT ORGANIZATION AND FORMATTING:

1) Text: Reports must have a single 'Results and Discussion' section.

2) Figure formatting: Reports may have up to 5 main text figures. Scale bars must be present on all microscopy images, including inset magnifications. Molecular weight or nucleic acid size markers must be included on all gel electrophoresis. Also, please avoid pairing red and green for images and graphs to ensure legibility for color-blind readers. If red and green are paired for images, please ensure that the particular red and green hues used in micrographs are distinctive with any of the colorblind types. If not, please modify colors accordingly or provide separate images of the individual channels.

3) Statistical analysis: Error bars on graphic representations of numerical data must be clearly described in the figure legend. The number of independent data points (n) represented in a graph must be indicated in the legend. Please indicate whether 'n' refers to technical or biological replicates (i.e. number of analyzed cells, samples or animals, number of independent experiments). If independent experiments with multiple biological replicates have been performed, we recommend using distribution-reproducibility SuperPlots (please see Lord et al., JCB 2020) to better display the distribution of the entire dataset, and report statistics (such as means, error bars, and P values) that address the reproducibility of the findings.

Statistical methods should be explained in full in the materials and methods. For figures presenting pooled data the statistical measure should be defined in the figure legends. Please also be sure to indicate the statistical tests used in each of your experiments (both in the figure legend itself and in a separate methods section) as well as the parameters of the test (for example, if you ran a t-test, please indicate if it was one- or two-sided, etc.). Also, if you used parametric tests, please indicate if the data distribution was tested for normality (and if so, how). If not, you must state something to the effect that "Data distribution was assumed to be normal but this was not formally tested."

4) Materials and methods: Should be comprehensive and not simply reference a previous publication for details on how an experiment was performed. Please provide full descriptions (at least in brief) in the text for readers who may not have access to referenced manuscripts. The text should not refer to methods "...as previously described." Please also indicate the acquisition and quantification methods for immunoblotting/western blots.

5) For all cell lines, vectors, strains, constructs/cDNAs, etc. - all genetic material: please include database / vendor ID (e.g. Addgene, ATCC, etc.) or if unavailable, please briefly describe their basic genetic features, even if described in other published work or gifted to you by other investigators (and provide references where appropriate). Please be sure to provide the sequences for all of your oligos: primers, si/shRNA, RNAi, gRNAs, etc. in the materials and methods. You must also indicate in the methods the source, species, and catalog numbers/vendor identifiers (where appropriate) for all of your antibodies, including secondary. If antibodies are not commercial, please add a reference citation if possible.

6) Microscope image acquisition: The following information must be provided about the acquisition and processing of images:

- a. Make and model of microscope
- b. Type, magnification, and numerical aperture of the objective lenses
- c. Temperature
- d. Imaging medium
- e. Fluorochromes
- f. Camera make and model

g. Acquisition software

h. Any software used for image processing subsequent to data acquisition. Please include details and types of operations involved (e.g., type of deconvolution, 3D reconstitutions, surface or volume rendering, gamma adjustments, etc.).

7) References: There is no limit to the number of references cited in a manuscript. References should be cited parenthetically in the text by author and year of publication. Abbreviate the names of journals according to PubMed.

8) Supplemental materials: Reports generally may have up to 5 supplemental figures and 10 videos. You currently exceed this limit but, in this case, we will be able to give you the extra space. Tables, like figures, should be provided as individual, editable files. A summary of all supplemental material should appear at the end of the Materials and methods section. Please include one brief sentence per item.

9) Video legends: Should describe what is being shown, the cell type or tissue being viewed (including relevant cell treatments, concentration and duration, or transfection), the imaging method (e.g., time-lapse epifluorescence microscopy), what each color represents, how often frames were collected, the frames/second display rate, and the number of any figure that has related video stills or images.

10) eTOC summary: A ~40-50 word summary that describes the context and significance of the findings for a general readership should be included on the title page. The statement should be written in the present tense and refer to the work in the third person. It should begin with "First author name(s) et al..." to match our preferred style.

11) Conflict of interest statement: JCB requires inclusion of a statement in the acknowledgements regarding competing financial interests. If no competing financial interests exist, please include the following statement: "The authors declare no competing financial interests." If competing interests are declared, please follow your statement of these competing interests with the following statement: "The authors declare no further competing financial interests."

12) A separate author contribution section is required following the Acknowledgments in all research manuscripts. All authors should be mentioned and designated by their first and middle initials and full surnames. We encourage use of the CRediT nomenclature (<https://casrai.org/credit/>).

13) ORCID IDs: ORCID IDs are unique identifiers allowing researchers to create a record of their various scholarly contributions in a single place. Please note that ORCID IDs are required for all authors. At resubmission of your final files, please be sure to provide your ORCID ID and those of all co-authors.

14) JCB requires authors to submit Source Data used to generate figures containing gels and Western blots with all revised manuscripts. This Source Data consists of fully uncropped and unprocessed images for each gel/blot displayed in the main and supplemental figures. For assays performed using capillary electrophoresis and/or immunoassay-based detection, authors should instead provide the electropherogram graph(s) for each experiment, plotting fluorescence/chemiluminescence intensity vs. molecular weight/size. Since your paper includes cropped gel and/or blot images, please be sure to provide one Source Data file for each figure gels, blots, and/or capillary electrophoresis assays along with your revised manuscript files. File names for Source Data figures should be alphanumeric without any spaces or special characters (i.e., SourceDataF#, where F# refers to the associated main figure number or SourceDataFS# for those associated with Supplementary figures). For traditional gels and blots, the lanes of the gels/blots should be labeled as they are in the associated figure, the place where cropping was applied should be marked (with a box), and molecular weight/size standards should be labeled wherever possible. For capillary electrophoresis assays, each trace in the graph should be color-coded and labeled to indicate which protein, gene, or sample is being measured (please try to avoid red/green combinations to accommodate our color-blind readers).

Source Data files will be directly linked to specific figures in the published article. Source Data Figures should be provided as individual PDF files (one file per figure). Authors should endeavor to retain a minimum resolution of 300 dpi or pixels per inch. Please review our instructions for export from Photoshop, Illustrator, and PowerPoint here: <https://rupress.org/jcb/pages/submission-guidelines#revised>

15) Journal of Cell Biology now requires a data availability statement for all research article submissions. These statements will be published in the article directly above the Acknowledgments. The statement should address all data underlying the research presented in the manuscript. Please visit the JCB instructions for authors for guidelines and examples of statements at (<https://rupress.org/jcb/pages/editorial-policies#data-availability-statement>).

B. FINAL FILES:

Thank you for your attention to these final processing requirements. Please contact the journal office with any questions at cellbio@rockefeller.edu.

Thank you for this interesting contribution, we look forward to publishing your paper in Journal of Cell Biology.

Sincerely,

Hong Zhang, PhD
Monitoring Editor
Journal of Cell Biology

Dan Simon, PhD
Scientific Editor
Journal of Cell Biology